# *Tiller Number1* encodes an ankyrin repeat protein that controls tillering in bread wheat

Chunhao Dong[1,5], Lichao Zhang[1,5], Qiang Zhang[1,2], Yuxin Yang[1], Danping Li[1], Zhencheng Xie[1], Guoqing Cui[1], Yaoyu Chen[1], Lifen Wu[1], Zhan Li[1], Guoxiang Liu[1], Xueying Zhang[1], Cuimei Liu[3], Jinfang Chu[3,4], Guangyao Zhao[1], Chuan Xia[1], Jizeng Jia ®[1], Jiaqiang Sun ®[1] ✉, Xiuying Kong ®[1] ✉ & Xu Liu ®[1] ✉

Wheat (*Triticum aestivum* L.) is a major staple food for more than one-third of the world's population. Tiller number is an important agronomic trait in wheat, but only few related genes have been cloned. Here, we isolate a wheat mutant, *tiller number1* (*tn1*), with much fewer tillers. We clone the *TN1* gene via map-based cloning: *TN1* encodes an ankyrin repeat protein with a transmembrane domain (ANK-TM). We show that a single amino acid substitution in the third conserved ankyrin repeat domain causes the decreased tiller number of *tn1* mutant plants. Resequencing and haplotype analysis indicate that *TN1* is conserved in wheat landraces and modern cultivars. Further, we reveal that the expression level of the abscisic acid (ABA) biosynthetic gene *TaNCED3* and ABA content are significantly increased in the shoot base and tiller bud of the *tn1* mutants; TN1 but not tn1 could inhibit the binding of TaPYL to TaPP2C via direct interaction with TaPYL. Taken together, we clone a key wheat tiller number regulatory gene *TN1*, which promotes tiller bud outgrowth probably through inhibiting ABA biosynthesis and signaling.

Wheat (*Triticum aestivum* L.) is a major staple food for one-third of the human race and supplies ~20% of daily calorie intake for humans[1]. Wheat production accounts for ~30% of global cereal crops[2]. To feed the growing world population, it is estimated that wheat yield needs to increase from the current 3 tons per hectare to 5 tons per hectare[3,4]. Hence, it is important to dissect the genetic basis for wheat yield.

Tiller number is a major agronomic trait in wheat. Although several quantitative trait loci (QTL) and genes that regulate tiller number have been mapped in wheat[5,6], only *tiller inhibition* (*tin*), which encodes a cellulose-synthase-like (Csl) protein, has been cloned through map-based cloning approach[7–9]. The *tin2* gene was mapped to chromosome 2A in bread wheat[10], while the *tin3* gene was mapped to the distal region of chromosome arm 3AᵐL of diploid wheat (*Triticum monococcum*)[11,12].

The *ftin* gene regulates fertile tiller development in 'Pubing3558' (a derivative of a wide cross between the common wheat cv. 'Fukuhoko-mugi' and wild grass [*Agropyron cristatum* L.]) and was mapped to chromosome 1AS[13]. In addition, several stable major QTL that modulate wheat tillering have been identified. For example, *QHt.nau-2D* and *Qltn.sicau-2D* were mapped to chromosome 2D and explain 27.2–40.4% and 19.1% of the phenotypic variance, respectively[14,15]. The *QPtn.sau-4B* was mapped to a 0.55-cM interval on chromosome 4B, containing the *TraesCS4B02G042700* gene that encodes the ortholog of maize (*Zea mays* L.) TEOSINTE BRANCHED 1 (TB1)[16].

The phytohormones auxin, cytokinin (CK), and strigolactone (SL) play important roles in regulating tillering and branching[17,18]. Recent studies have reported that the phytohormone abscisic acid (ABA) also acts as a negative regulator of tillering and branching in rice

[1]State Key Laboratory of Crop Gene Resources and Breeding, Institute of Crop Sciences, Chinese Academy of Agricultural Sciences, Beijing 100081, China. [2]State Key Lab of Rice Biology, China National Rice Research Institute, Hangzhou 310006, China. [3]National Centre for Plant Gene Research (Beijing), Innovation Academy for Seed Design, Institute of Genetics and Developmental Biology, Chinese Academy of Sciences, Beijing 100101, China. [4]College of Advanced Agricultural Sciences, University of Chinese Academy of Sciences, Beijing 100049, China. [5]These authors contributed equally: Chunhao Dong, Lichao Zhang. ✉e-mail: sunjiaqiang@caas.cn; kongxiuying@caas.cn; liuxu03@caas.cn

(*Oryza sativa*)[19], Arabidopsis (*Arabidopsis thaliana*)[20,21], cotton (*Gossypium hirsutum*)[22], and bean (*Phaseolus vulgaris*)[23]. In Arabidopsis, the branching repressor BRANCHED1 (BRC1), the ortholog of maize TB1, directly activates the expression of the key ABA biosynthesis gene *9-CIS-EPOXICAROTENOID DIOXIGENASE 3* (*NCED3*), thereby enhancing ABA accumulation in axillary buds and consequently inhibiting axillary bud outgrowth[21].

In this work, we isolate a wheat mutant, *tiller number1* (*tn1*), with a dramatic reduction in tiller number. Via map-based cloning, we clone the *TN1* gene, which encodes an ankyrin repeat protein. We demonstrate here that the inhibition of tiller bud outgrowth in the *tn1* mutant may be caused by enhanced ABA accumulation and ABA signaling in the tiller buds.

## Results

### Characterization of the *tn1* mutant

We characterized a low-tillering mutant, *tn1*, from an ethyl methanesulfonate (EMS)-mutagenized population of the wheat variety 'Yanzhan4110' (YZ4110). At the third-leaf stage, we observed tillers on the unelongated basal internodes of the parental line YZ4110, whereas no tillers appeared from the leaf axils of the *tn1* mutant (Fig. 1a). At the jointing stage, 15–24 tillers emerged on the basal part of YZ4110 plants, in contrast to only 1–4 tillers were observed in the *tn1* mutant (Fig. 1b, d). At the grain-filling stage, YZ4110 plants produced more than 15 tillers, but only 1–4 tillers developed in the *tn1* mutant (Fig. 1c, e). In addition, plant height, spike length, spikelet number and grain number per spike in the *tn1* mutant were significantly lower compared to YZ4110 (Supplementary Fig. 1a–d).

Tiller development consists of two stages: tiller bud initiation and tiller bud outgrowth[24,25]. To determine which stage of tiller bud development was affected in the *tn1* mutant, we conducted anatomical and histological observations. We determined that tiller buds can still form in the *tn1* mutant, but most of these tiller buds failed to outgrow in subsequent development (Fig. 1f, g). These observations suggest

that the low-tillering phenotype of the *tn1* mutant can be attributed to the suppression of tiller bud outgrowth.

### Map-based cloning of *TN1*

We performed a genetic analysis of the *tn1* mutant by characterizing the phenotype of its progeny. In a genetic cross between YZ4110 and the *tn1* mutant, all $F_1$ plants had the same number of tillers as YZ4110 plants (Supplementary Fig. 2a), indicating that *tn1* is a recessive mutation. Tiller number in 431 $F_2$ individuals followed a bimodal distribution with a gap at four tillers, which was considered as the trait value boundary (Supplementary Fig. 2b). Accordingly, we divided the $F_2$ population into low-tillering plants (with one to four tillers) and multi-tillering plants (with more than four tillers). The segregation ratio of these two groups fits a mendelian model of 3:1 (315 multi-tillering plants:116 low-tillering plants; $\chi^2 = 0.842 < \chi^2_{0.05,1} = 3.84$) (Supplementary Table 1). These results indicate that the low-tillering phenotype of the *tn1* mutant is controlled by a single recessive nuclear gene.

To clone the *TN1* gene, we crossed the *tn1* mutant to the free-tillering variety 'Jimai20' (JM20). The resulting 46 $F_1$ individuals showed multi-tillering phenotypes (Supplementary Fig. 2c, d, Supplementary Table 1). By selfing $F_1$ plants, we obtained a biparental $F_2$ mapping population comprising 11,155 individuals, of which 2952 had one to four tillers and were used for gene mapping. We first conducted bulked segregant analysis and wheat 660 K single-nucleotide polymorphism (SNP) array-based genotyping using the individuals with extreme phenotypes from the $F_{2:3}$ progenies. A total of 1058 SNPs were polymorphic, of which 169 were located on chromosome 6B (Supplementary Fig. 3a), indicating that the *TN1* gene is most likely located on chromosome 6B. Further analysis showed that 92 SNPs are enriched over a 2- to 11-Mb interval at the distal end of the short arm of chromosome 6B (Supplementary Fig. 3b). We next selected 35 SNPs in this interval to design kompetitive allele-specific PCR (KASP) markers, of which we tested the eight that showed polymorphisms further in 186 recessive individuals. We thus narrowed down the *TN1* interval to a

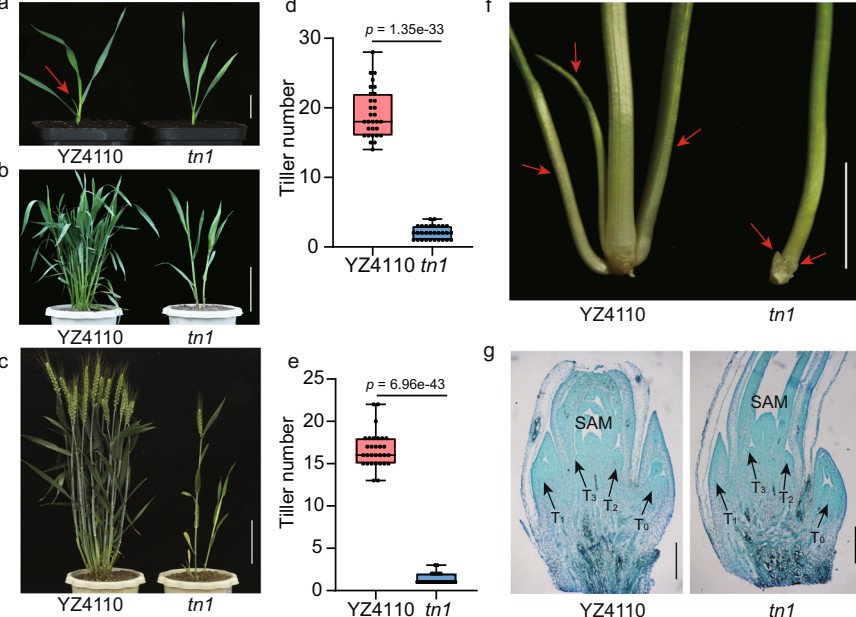

**Fig. 1 | Phenotypes of the *tn1* mutant. a–c** Morphology of YZ4110 and the *tn1* mutant at the third-leaf stage (**a**), jointing stage (**b**), and grain-filling stage (**c**). The red arrow in **a** indicates the position of the tiller bud. Scale bars, 2.5 cm in **a**, 15 cm in **b** and **c**. **d**, **e** Tiller number in YZ4110 and *tn1* at the jointing stage (**d**) and grain-filling stage (**e**). Data are means ± SD (*n* = 30 biologically independent samples), and *p* values are indicated by two-tailed unpaired *t*-test. The two whiskers of the box plot and the middle, upper, and lower box lines represent the maximum, minimum,

median and two quartiles of values in each group. **f** Micrograph of tiller buds at the third-leaf stage in YZ4110 and *tn1*. Red arrows indicate the positions of tiller buds. Scale bar, 1 cm. **g** Microscopy observations of longitudinal sections of the axillary buds. Images are representative of two independent experiments. Black arrows indicate the positions of axillary buds. $T_0$–$T_3$ represent the axillary buds that arose from the coleoptile and the first-, second-, and third-leaf axils, respectively. Scale bars, 0.5 mm. Source data are provided as a Source Data file.

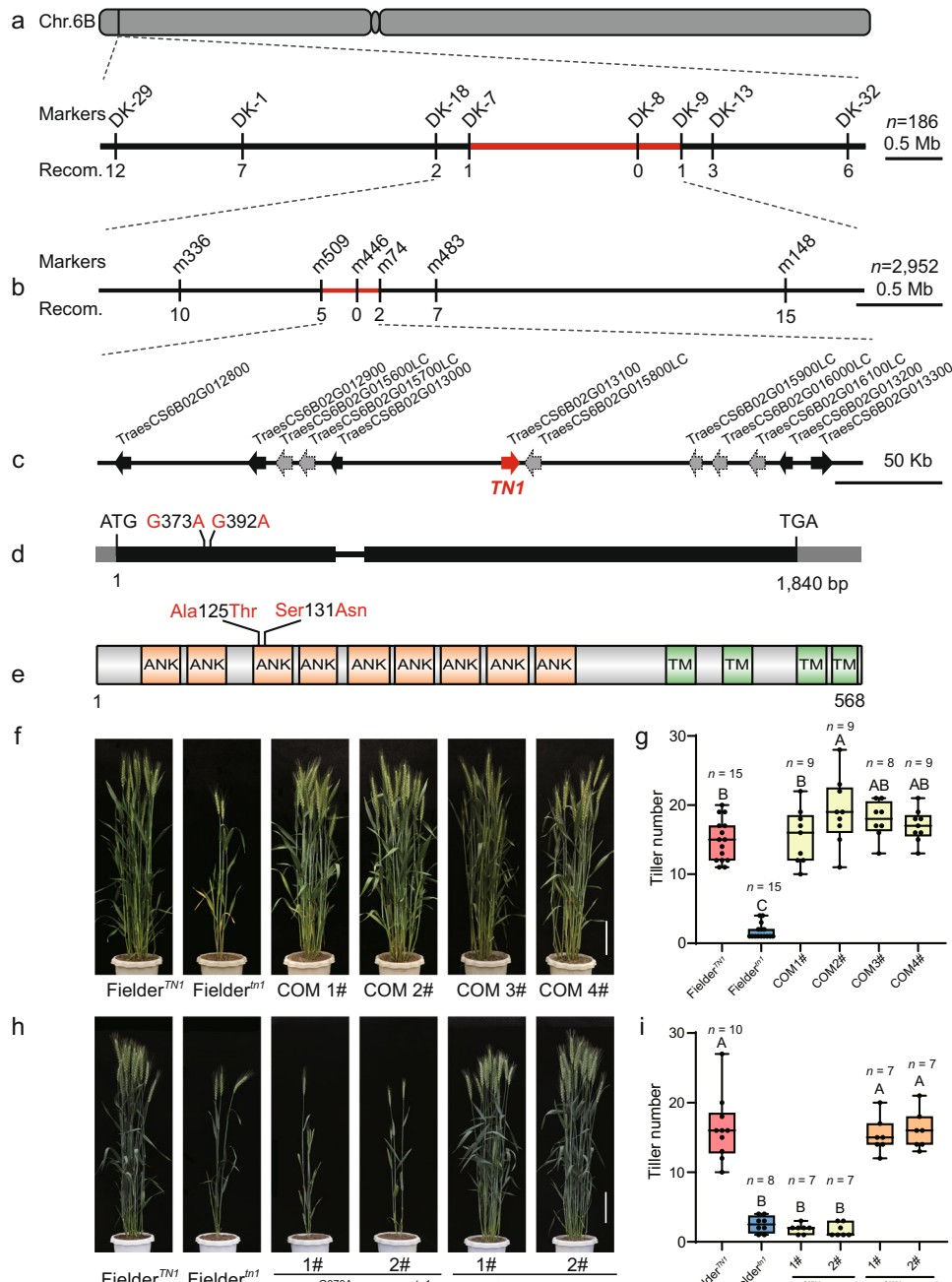

**Fig. 2 | Map-based cloning of *TN1*. a**, **b** Positional cloning of *TN1*. **a** Using 186 low-tillering individuals, *TN1* was first mapped to an ~2.3-Mb genomic DNA region between KASP markers DK-7 and DK-9, with no recombinants detected between *tn1* and the marker DK-8. **b** By using 2952 recessive individuals, *TN1* was fine-mapped to an ~352-kb genomic region flanked with SSR markers m509 and m74, with no recombinants detected between *tn1* and marker m446. The numbers below the lines indicate the number of recombinants. **c** Genes annotated within the candidate region. The black arrows and dashed gray arrows represent the high-confidence (HC) and low-confidence (LC) genes, respectively. The red arrow indicates *TN1*. **d** Schematic diagram of the *TN1* locus and mutation sites. Black boxes, exons; gray boxes, untranslated regions (UTRs); black horizontal line, intron. The nucleotide changes and positions of mutations are shown as indicated. **e** Schematic diagram of the predicted TN1 protein. ANK, ankyrin repeat domain; TM, transmembrane helix region. The amino acids changes are shown as indicated. **f**, **g** Morphology (**f**) and tiller number (**g**) for Fielder^{TN1}, Fielder^{tn1}, and positive transgenic lines (COM 1#–4#) from the complementation test. **h**, **i** Morphology (**h**) and tiller number (**i**) of the transgenic lines with the *TN1pro:TN1^{G373A}* or *TN1pro:TN1^{G392A}* construct. Scale bar, 20 cm in **f** and **h**. Data in **g** and **i** are means ± SEM, and tiller numbers were measured at grain-filling stage from independent transgenic plants. Different letters indicate significant differences between groups, as determined by one way ANOVA with Duncan's multiple range tests ($p < 0.01$). The two whiskers of the box plot and the middle, upper, and lower box lines represent the maximum, minimum, median and two quartiles of values in each group. Source data are provided as a Source Data file.

2.3-Mb region flanked by markers DK-7 and DK-9, with marker DK-8 showing complete linkage to *tn1* (Fig. 2a, Supplementary Fig. 4, Supplementary Data 1).

To further reduce the region of the *TN1* gene, we designed simple sequence repeat (SSR) markers based on the sequence of

IWGSC RefSeq v0.4 with BatchPrimer3[26] to genotype a larger segregating population consisting of 2952 low-tillering $F_2$ individuals. We fine-mapped *TN1* to a 352-kb region flanked by markers m509 and m74, with complete co-segregation with marker m446 (Fig. 2b, Supplementary Data 1). The 352-kb region contains 12 putative

genes according to the IWGSC RefSeq Annotation v1.1[27], with six high-confidence genes and six low-confidence genes (Fig. 2c, Supplementary Table 2). We amplified and sequenced these 12 genes using specific primers from the YZ4110 and *tn1* genomes (Supplementary Data 2). We detected two SNPs (SNP$^{G373A}$ and SNP$^{G392A}$) in the first exon of *TraesCS6B02G013100*, resulting in the amino acid substitutions Ala-125-Thr and Ser-131-Asn (Fig. 2d, e). *TraesCS6B02G013100* encodes an ankyrin repeat (ANK) family protein containing nine ANK domains and four transmembrane domains, and the two SNPs were located in the region encoding the third ANK motif.

To determine whether the two SNPs in *TraesCS6B02G013100* co-segregated with the low-tillering phenotype, we generated 275 individuals of the BC$_2$F$_2$ population by crossing YZ4110 and the *tn1* mutant. Of these 275 individuals, 73 low-tillering individuals contained the same mutations as the *tn1* mutant, while 61 multi-tillering individuals showed the same genotype as YZ4110, and the remaining 141 multi-tillering individuals were heterozygous lines. The results strongly suggested that *TraesCS6B02G013100* is the candidate gene for *TN1*.

### Functional validation of *TN1* gene

To confirm that *TraesCS6B02G013100* is responsible for the low-tillering phenotype of the *tn1* mutant, we conducted a genetic complementation test. Because the *tn1* mutant is in the YZ4110 background, which is difficult to transform, we generated a low-tillering Fielder$^{tn1}$ line from a BC$_4$F$_3$ population derived from a genetic cross between *tn1* and 'Fielder'. We then cloned a *TraesCS6B02G013100* genomic fragment containing a 3251-bp promoter region upstream from the start codon, a 1840-bp coding region, and a 2266-bp terminator region downstream from the stop codon from the YZ4110 genomic DNA into a plant binary vector. We transformed the resulting construct into Fielder$^{tn1}$ by Agrobacterium (*Agrobacterium tumefaciens*)−mediated transformation. We also evaluated the activity of the 3251-bp promoter fragment using green fluorescent protein (GFP) as a reporter gene (Supplementary Fig. 5). The transgenic lines restored the tillering phenotype of the Fielder$^{tn1}$ background to that of Fielder$^{TN1}$, demonstrating that the *TraesCS6B02G013100* gene can rescue the low-tillering phenotype of the *tn1* mutant (Fig. 2f, g).

Next, we asked which of the two SNPs (SNP$^{G373A}$ and SNP$^{G392A}$) in *TraesCS6B02G013100* gene is responsible for the low-tillering phenotype. To this end, we generated two constructs, each containing the *TraesCS6B02G013100* gene with only one SNP, and introduced them into Fielder$^{tn1}$, respectively. The construct with SNP$^{G392A}$ rescued the low-tillering phenotype in transgenic plants, indicating that SNP$^{G392A}$ does not affect the function of the protein encoded by *TraesCS6B02G013100* (Fig. 2h, i). By contrast, a construct harboring SNP$^{G373A}$ failed to rescue the low-tillering phenotype, indicating that SNP$^{G373A}$ is the causal mutation for this phenotype (Fig. 2h, i).

To further evaluate the role of *TraesCS6B02G013100* in regulating tillering, we isolated knockout mutants by the clustered regularly interspaced short palindromic repeats (CRISPR)/CRISPR-associated protein 9 (Cas9)−mediated genome editing. We designed two single guide RNAs (sgRNAs) to specifically and separately target the 5′ untranslated region (UTR) and the first exon of *TraesCS6B02G013100* and then introduced them into the pBUE413 vector and transformed them into Fielder (Fig. 3a, b). We obtained 15 transgenic plants, four of which had homozygous or biallelic mutations at the target sites and were used for phenotypic analysis. In the T$_1$ generation, these mutants displayed reduced tiller numbers similar to those seen for the *tn1* mutant (Fig. 3c−e).

In summary, these results support the conclusion that *TraesCS6B02G013100* is the *TN1* gene and that SNP$^{G373A}$ causes the low-tillering phenotype.

### *TN1* encodes a transmembrane ankyrin repeat protein

The *TN1* gene encodes an ANK family protein containing nine ANK domains and four transmembrane domains (Fig. 2e). The N-terminal region of TN1 contains nine ANK domains based on the SMART protein prediction program (http://smart.embl-heidelberg.de/), and the C-terminal region contains four predicted transmembrane helices according to a hidden Markov model (TMHMM Server v. 2.0) (Fig. 2e, Supplementary Fig. 6). In addition, the Ala-125-Thr substitution occurring in the tn1 protein is predicted to cause the extension of the α-helix in TN1, based on the software I-TASSER (https://zhanglab.ccmb.med.umich.edu/I-TASSER/) (Supplementary Fig. 7).

To confirm these predictions experimentally, we co-transfected wheat mesophyll protoplasts with the *35Spro:TN1-GFP* construct and *35Spro:PIP2-mCherry*, encoding a plasma membrane and endoplasmic reticulum marker[28]. TN1-GFP co-localized with PIP2-mCherry (Fig. 4a), demonstrating that TN1 is a membrane-located protein.

### *TN1* is mainly expressed in shoot apical and axillary meristems

To better understand TN1 function, we investigated the temporal and spatial expression patterns of *TN1*. Reverse transcription quantitative PCR (RT-qPCR) analysis determined that *TN1* is highly expressed in tiller buds (Fig. 4b). RNA in situ hybridization further revealed that *TN1* is mainly expressed in shoot apical meristems and axillary meristems and extended to the entire tiller buds (Fig. 4c−f). Although *TN1* is also highly expressed in young spikes, no branching defect was observed in the spikes of *tn1* mutant at different spike developmental stages (Supplementary Fig. 8).

### Phylogenetic analysis of *TN1* homologs

Bread wheat is a hexaploid species harboring three subgenomes: A, B and D. To identify the homologs of *TN1* in hexaploid wheat, we performed a BLAST search using the complete *TN1* coding sequence as a query. We identified three ANK proteins, encoded by *TraesCS6B02G437600*, *TraesCS6B02G013000* and *TraesCS6D02G011400*, sharing 88.7%, 75.5%, and 74.3% sequence identity with *TN1*, which were designated as *TN1-like-6BL*, *TN1-like-6BS*, and *TN1-like-6DS*, respectively (Supplementary Fig. 9). A phylogenetic analysis showed that *TN1* and *TN1-like-6BL* cluster together, whereas *TN1-like-6BS* and *TN1-like-6DS* belong to a separate subclade (Supplementary Fig. 10a). We further analyzed the expression levels of *TN1* and *TN1-like-6BL* at the shoot base and tiller buds of YZ4110. *TN1* was highly expressed at both the shoot base and in tiller buds, whereas *TN1-like-6BL* was barely expressed (Supplementary Fig. 10b), suggesting that *TN1* may play a predominant role in regulating tiller development.

To determine whether Ala-125 and Ser-131 in TN1 are conserved in land plants, we aligned the polypeptide sequence of the third ANK domain (33 amino acids) in TN1 and its homologs for phylogenetic analysis (Supplementary Data 3). We observed that the 24 aligned sequences cluster into two groups following the monocot/dicot divide, suggestive of the evolutionary divergence of TN1 (Supplementary Fig. 11a). Protein sequence analysis showed that the residue Ala-125 is highly conserved in land plants (Supplementary Fig. 11a, b). These observations suggest that the conserved Ala-125 amino acid may be important for TN1 function.

Next, we amplified and sequenced *TN1* from the Chinese mini-core wheat collection to investigate natural variation in this gene (Supplementary Data 4)[29]. The sequence of *TN1* was identical in all cultivars tested. We also detected no polymorphism in another 145 wheat cultivars, which were re-sequenced with an average 17.94× read depth[30], or in the 890 diverse wheat landraces and cultivars (http://wheatgenomics.plantpath.ksu.edu/1000EC/)[31]. These results suggest that the function of *TN1* may be necessary to maintain proper tiller number in wheat.

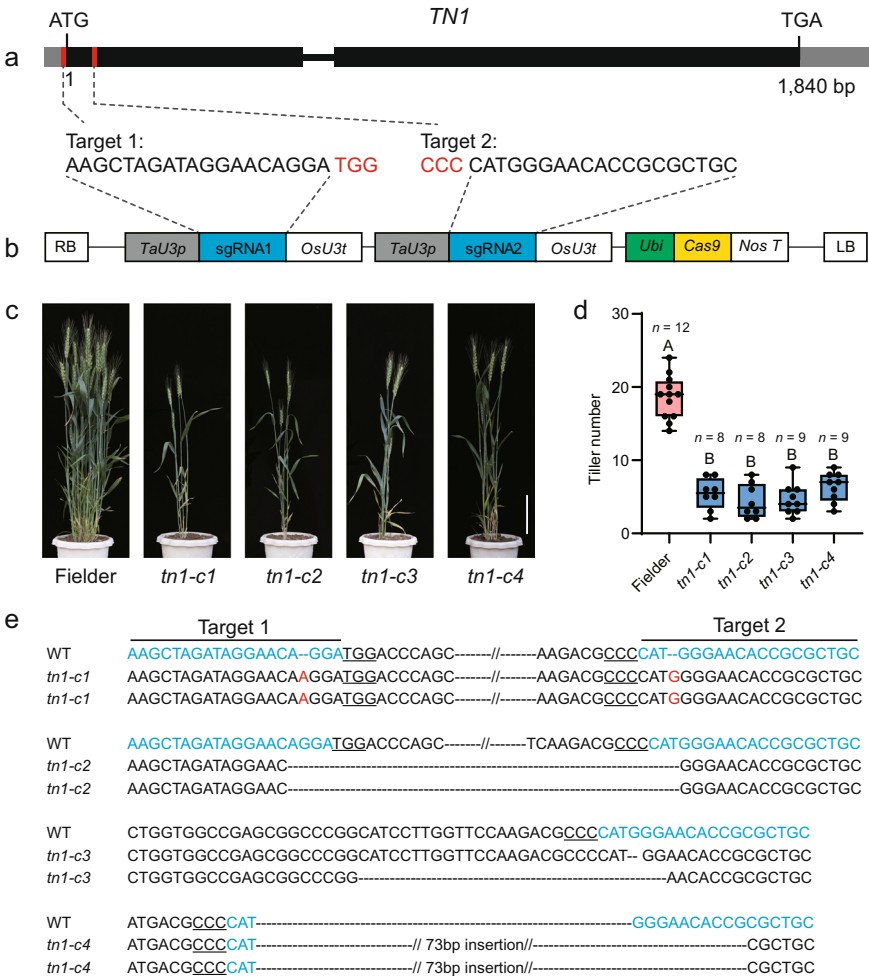

**Fig. 3 | The *tn1* mutants generated by CRISPR/Cas9 exhibit reduced tiller numbers. a** Schematic diagram of *TN1* and the target sites for the single guide RNAs. Red boxes in the gene model represent the target sites, and sequences are shown underneath with the protospacer-adjacent motif (PAM) highlighted in red. **b** Schematic diagram of the gene-editing construct used for transformation. *TaU3p*, wheat *U3* promoter; sgRNA, single guide RNA; *OsU3t*, rice *U3* terminator; *Ubi*, *Zea mays Ubiquitin* promoter; *NosT*, *Nos* terminator. LB/RB, left/right border. **c**, **d** Representative phenotypes (**c**) and tiller number (**d**) of the *tn1* mutants generated by CRISPR/Cas9-mediated gene editing. Scale bar, 20 cm in **c**. Data in **d** are means ± SEM, and tiller numbers were measured at grain-filling stage from independent transgenic plants. Different letters indicate significant differences between groups, as determined by one way ANOVA with Duncan's multiple range tests ($p < 0.01$). The two whiskers of the box plot and the middle, upper, and lower box lines represent the maximum, minimum, median and two quartiles of values in each group. **e** Genotype of the mutant lines. The target sites are marked in blue, and the PAM is underlined. The red letters indicate nucleotide insertions and the black dashed lines represent nucleotide deletions. Source data are provided as a Source Data file.

## TN1 regulates tillering through the ABA pathway

To understand the molecular basis of TN1-regulated tillering in wheat, we performed a transcriptome sequencing (RNA-seq) analysis using the shoot base and tiller buds from YZ4110 and the *tn1* mutant (Fig. 5a, b). We identified 5372 and 4672 differentially expressed genes (DEGs) between YZ4110 and the *tn1* mutant in the shoot base and tiller buds, respectively (Supplementary Data 5, 6). Gene Ontology (GO) analysis for the 1876 DEGs common to the shoot base and tiller buds revealed a significant enrichment in phytohormone-mediated signaling pathways, phytohormone metabolic/biosynthetic process, shoot system development, response to abscisic acid, and abscisic acid metabolic process (Supplementary Fig. 12a, b, Supplementary Data 7). For the molecular function category, the DEGs were enriched in DNA binding and transcription factor activity, as illustrated by the overrepresentation of genes belonging to the transcription factor families, such as bHLH, bZIP, ERF, MADS-box, MYB, and WRKY. For cellular components, GO analysis showed a significant enrichment for the membrane protein complex, intracellular part, and trans-Golgi network membrane categories (Fig. 5c, Supplementary Fig. 12b, Supplementary Data 7).

The *TB1/BRC1* genes play important roles in regulating tillering and branching in plants[18,32–34]. However, our RNA-seq and RT-qPCR analysis showed that the expression levels of *TaTB1* are comparable between YZ4110 and the *tn1* mutant (Supplementary Fig. 13). Notably, several ABA-related genes were enriched among the DEGs, including genes related to ABA biosynthesis and signaling (Fig. 5c). Of these, the key ABA biosynthetic genes *TraesCS5B02G029300* (*TaNCED3-5B*) and *TraesCS5D02G038800* (*TaNCED3-5D*) were significantly upregulated in the shoot base and in the tiller buds of the *tn1* mutant, respectively, based on our RNA-seq data (Fig. 5c, Supplementary Data 5, 6). RT-qPCR analysis confirmed that the transcript levels of *TaNCED3-5B* and *TaNCED3-5D* are significantly higher in the *tn1* mutant relative to the wild type (Fig. 5d). Moreover, a number of ABA signaling–related genes, including *PYR-LIKE* (*PYL*), *PROTEIN PHOSPHATASE 2C* (*PP2C*), and *SNF1-REGULATED PROTEIN KINASE2* (*SnRK2*), were also significantly upregulated in the *tn1* mutant (Supplementary Data 5, 6, Supplementary Fig. 14).

A previous study reported that ZmbZIP4 directly activates the transcription of *ZmNCED3* in maize[35]. Importantly, we noticed that the

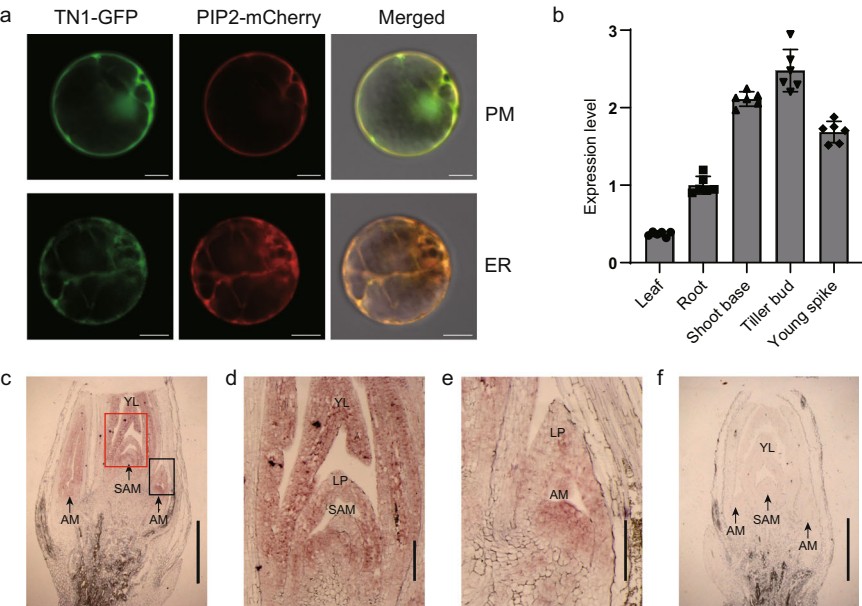

**Fig. 4 | Subcellular localization and expression pattern analyses. a** Subcellular localization of TN1 and co-localization with PIP2-mCherry in the plasma membrane (PM) pattern and endoplasmic reticulum (ER) network pattern. Images are representative of three independent experiments. Scale bars, 10 μm. **b** RT-qPCR analysis of relative *TN1* expression levels in various tissues of YZ4110. Leaves, roots, shoot bases, and tiller buds were collected at the third-leaf stage, and the young spikes were collected at the booting stage. The relative expression levels were normalized to *TaGAPDH*. Data are means ± SD of six biological replicates. **c–f** In situ hybridization analysis of *TN1* in shoot bases. Images are representative of two independent experiments. The regions in the red and black boxes in **c** are enlarged in **d** and **e**, respectively. SAM shoot apical meristem, AM axillary meristem, LP leaf primordium, YL young leaf. Sense probe control is shown in **f**. Scale bars, 1 mm in **c** and **f**; 0.2 mm in **d** and **e**. Source data are provided as a Source Data file.

expression levels of *TabZIP-5A/5B/5D*, the homologs of maize *ZmbZIP4*, are higher in the *tn1* mutant relative to the wild type (Supplementary Data 5, 6, Supplementary Fig. 15a). We further demonstrated that the transcription factors TabZIP-5A/5B/5D can activate transcription from the *TaNCED3* promoters in a transient transactivation system (Supplementary Fig. 15b–d).

Based on these observations, we predicted that ABA biosynthesis might be altered in the *tn1* mutant. To explore this idea, we measured endogenous ABA contents at the shoot base and in tiller bud of the *tn1* mutant. We observed that ABA levels in the shoot base of the *tn1* mutant were 30% higher compared to YZ4110; likewise, ABA levels were about two fold higher in the tiller bud of the *tn1* mutant compared to YZ4110 (Fig. 5e). In parallel, we measured indole-3-acetic acid (IAA) levels at the shoot base and in tiller bud of the *tn1* mutant. However, we detected no obvious difference in IAA content between *tn1* and YZ4110 (Fig. 5f).

ABA has been shown to play an important role in the inhibition of tillering and branching[19–22]. Indeed, the transgenic wheat lines overexpressing the ABA biosynthetic gene *TaNCED3-5D* had reduced tiller numbers compared the control Fielder (Supplementary Fig. 16a–c). Therefore, we predicted that the reduced tiller number of the *tn1* mutant might be caused by increased ABA contents in tiller buds (Fig. 5e). To test this idea, we first treated YZ4110 and the *tn1* mutant with different concentrations of exogenous ABA. We determined that tiller bud outgrowth of YZ4110 is greatly suppressed with as little as 5 μM exogenous ABA (Supplementary Fig. 17a). By contrast, treatment with the ABA biosynthesis inhibitor sodium tungstate partially rescued the low-tillering phenotype of the *tn1* mutant (Supplementary Fig. 17b). Together, these evidences indicate that the low-tillering phenotype of the *tn1* mutant may be, at least partially, caused by increased ABA levels in shoot base and tiller bud.

Further, we generated the mutants of *TaNCED3* genes in the Fielder^*tn1* background via CRISPR/Cas9-mediated gene editing. Using the coding sequence of *OsNCED3* (*LOC_Os03g44380*) as a query, we identified six homologous genes in the genome of Fielder[36]: *TaNCED3-5AS*

(*TraesFLD5A01G034600*), *TaNCED3-5AL* (*TraesFLD5A01G395800*), *TaNCED3-5BS* (*TraesFLD5B01G032700*), *TaNCED3-5BL* (*TraesFLD5B01G409000*), *TaNCED3-5DS* (*TraesFLD5D01G045900*), and *TaNCED3-5DL* (*TraesFLD5D01G418100*). Sequencing and phenotypic analyses showed that the sextuple mutants of *TaNCED3* genes in the T1 generation displayed partially rescued tiller numbers compared with Fielder^*tn1* (*Tanced3-c 2#–5#*, Supplementary Fig. 18a–c), while the triple mutant still showed similar tiller number as that of Fielder^*tn1* (*Tanced3-c 1#*, Supplementary Fig. 18a–c). These genetic evidences demonstrate that the low-tillering phenotype of the *tn1* mutant is actually related to the ABA pathway.

## TN1 interacts with TaPYL to inhibit its binding to TaPP2C

As the plasma membrane receptors of ABA, PYL proteins are essential for ABA perception and signal transduction in plants[37,38]. Considering that TN1 is a transmembrane protein, we wondered whether TN1 physically interacts with TaPYL-1D (TraesCS1D02G126900), which was significantly upregulated in the *tn1* mutant. To test this idea, we evaluated the physical association between TN1 and TaPYL-1D by firefly luciferase complementation imaging (LCI) assay in *N. benthamiana*. The results showed that the samples co-expressing cLUC-TN1 and nLUC-TaPYL-1D displayed strong luminescence signals (Fig. 6a). Further, we confirmed this interaction using the ankyrin repeat domain (TN1^ANK, residues 1–422) of TN1 by co-immunoprecipitation (Co-IP) system (Fig. 6b). Taken together, these results demonstrated that TN1 physically interacts with TaPYL.

Previous studies have well documented that PYLs directly bind PP2Cs to prevent its inhibition on the activity of SnRK2s, as a core signaling module of the ABA signaling pathway[37–39]. Here, we validated the physical interaction between TaPYL-1D and TaPP2C-7A (TraesCS7A02G241800) by LCI assay (Supplementary Fig. 19). To explore the effect of TN1 on the TaPYL function, we wondered whether TN1 affects the physical association between TaPYL-1D and TaPP2C-7A. To test this idea, we co-expressed TN1 with cLUC-TaPP2C-7A and nLUC-TaPYL-1D proteins in *N. benthamiana* leaves. As expected, the LUC intensities in

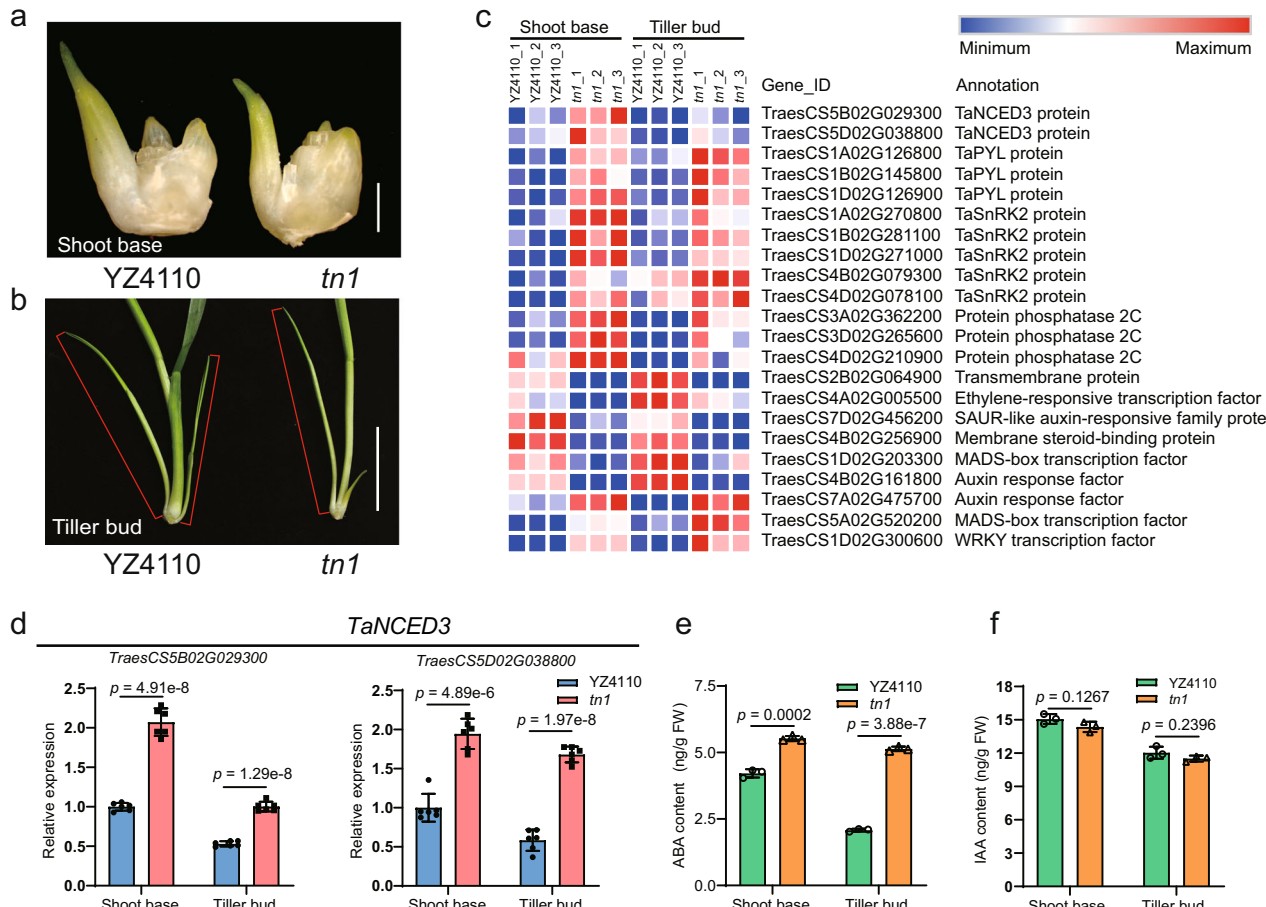

**Fig. 5 | TN1 affects the expression of abscisic acid (ABA) biosynthetic gene *TaNCED3* and ABA content in the shoot base and tiller bud. a, b** Morphology of the shoot base (**a**) and tiller buds (**b**) used for RNA-seq and ABA/IAA contents analysis. The tiller buds are indicated with red lines. Scale bars, 1 mm in (**a**), 2 cm in (**b**). **c** Heatmap representation of the expression levels of DEGs in the shoot base and tiller buds shared between YZ4110 and the *tn1* mutant. **d** RT-qPCR analysis showing the expression patterns of *TaNCED3* genes in YZ4110 and the *tn1* mutant.

*TaNCED3-5B, TraesCS5B02G029300*; *TaNCED3-5D, TraesCS5D02G038800*. The relative expression levels were normalized to *TaGAPDH*. Data are means ± SD of six biological replicates. **e, f** Measurement of ABA and indole-3-acetic acid (IAA) contents in the shoot base and tiller buds of the *tn1* mutant, respectively. Data are means ± SD of three biological replicates. The *p* values in **d–f** are indicated by two-tailed unpaired *t*-test. Source data are provided as a Source Data file.

cLUC-TaPP2C-7A/nLUC-TaPYL-1D/TN1-GFP co-expression samples (Fig. 6c, d, co-infiltration 1) were dramatically decreased by more than 50% compared to those in the cLUC-TaPP2C-7A/nLUC-TaPYL-1D/GFP EV samples (Fig. 6c, co-infiltration 3). Interestingly, no obvious differences were detected in LUC intensities between cLUC-TaPP2C-7A/nLUC-TaPYL-1D/tn1-GFP (Fig. 6c, d, co-infiltration 2) co-expression samples and cLUC-TaPP2C-7A/nLUC-TaPYL-1D/GFP EV samples (Fig. 6c, co-infiltration 3). Taken together, we propose that TN1 might participate in the ABA signaling pathway, at least partly, through inhibiting the physical association between TaPYL-1D and TaPP2C-7A (Fig. 6e).

## Discussion

Tiller number is a key element of wheat plant architecture that directly contributes to yield[40–42]. Here, we characterized a wheat mutant, *tn1*, with few tillers that was defective in tiller bud outgrowth (Fig. 1a–g). Through map-based cloning, we cloned the causal gene, *TN1*, which encodes an ANK-TM protein (Fig. 2e). Furthermore, we demonstrated that the reduced tiller number seen in the *tn1* mutant was caused by a single amino acid substitution (Ala-125-Thr) in the third conserved ANK domain (Fig. 2f–i). Natural variation analysis showed that the *TN1* gene is highly conserved in widely cultivated wheat varieties, indicating that the function of TN1 is necessary for wheat tillering (Supplementary Data 4).

In plants, ANK proteins are involved in protein–protein interactions and play a critical role in development and immunity[43–46]. Based on their domain compositions, ANK proteins can be classified into different subfamilies; among them, the ANK-TM subfamily containing the transmembrane domain (TM) is widely present in plants and participates in diverse cellular processes[47]. A total of 37 ANK-TM proteins have been identified in rice[48], 25 in tomato[49] and 40 in Arabidopsis[50]. *INCREASED TOLERANCE TO NACL1* (*ITN1*), encoding an ANK-TM subfamily member in Arabidopsis, is involved in salt tolerance and possibly in ABA signaling pathways[51]. *ACCELERATED CELL DEATH6* (*ACD6*), encoding a membrane protein with ANK domains, is a regulator of salicylic acid signaling in the Arabidopsis defense response[52,53]. The ANK-TM protein INEFFECTIVE GREENISH NODULES1 (IGN1) is required for the persistence of nitrogen-fixing symbiosis in root nodules in *Lotus japonicus*[54]. The ANK-TM protein Lr14a is involved in race-specific leaf rust disease resistance in wheat[55], but not in regulating wheat plant architecture. In this study, we clone the wheat tiller number regulatory gene *TN1*, which also encodes an ANK-TM protein. This study reveals that the ANK-TM subfamily protein TN1 plays an important role in regulating wheat tillering, providing new insight into the molecular basis of wheat plant architecture improvement.

Previous studies have shown that ABA plays an important role in the inhibition of tillering and branching bud outgrowth[19–22]. Exogenous ABA significantly inhibits the outgrowth of tiller buds in rice[19], wheat,

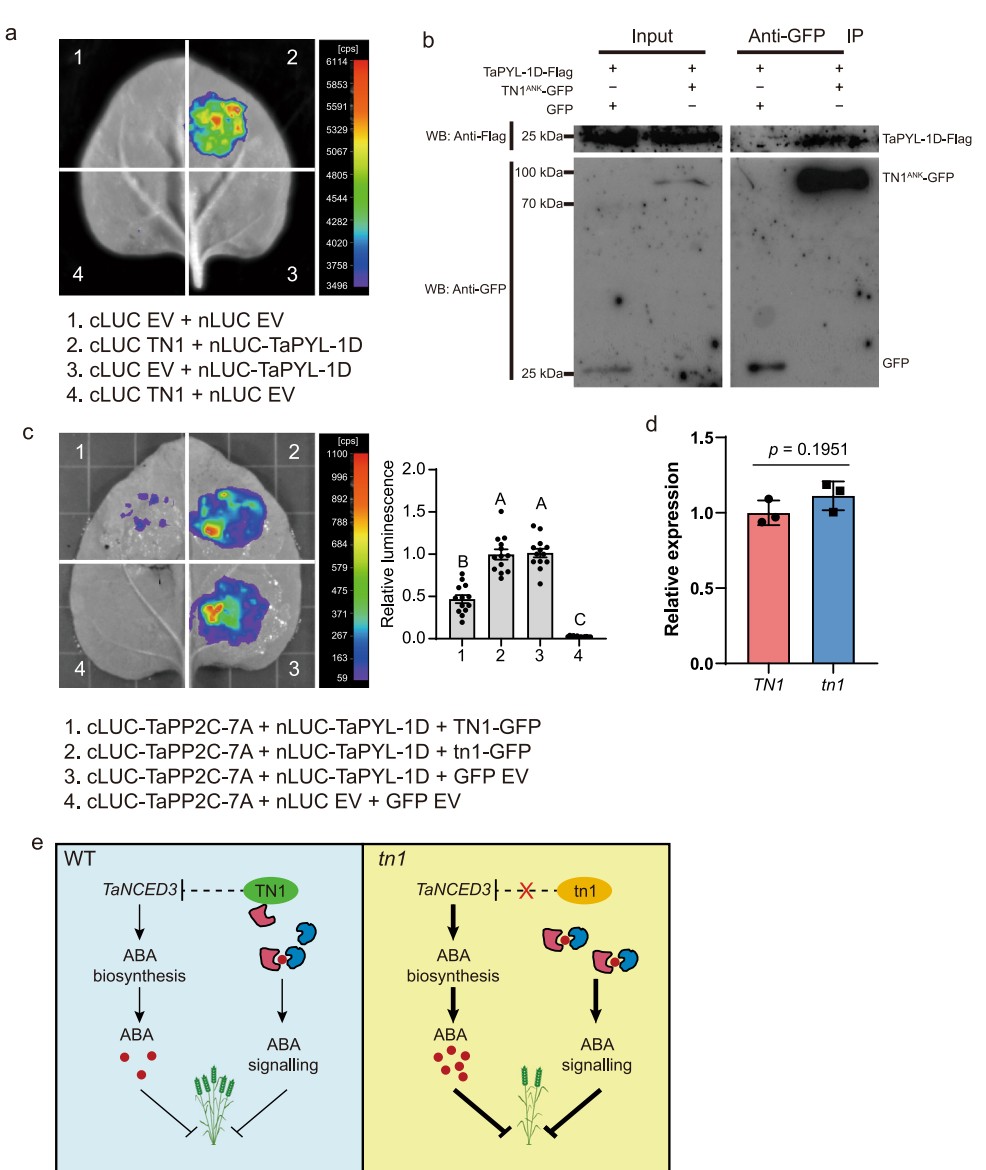

**Fig. 6 | TN1 interacts with TaPYL to inhibit the physical association between TaPYL and TaPP2C. a** Interaction between TN1 and TaPYL-1D revealed by LCI assay. **b** The ankyrin repeat domain (TN1^ANK, residues 1–422) of TN1 was used to test the interaction by co-immunoprecipitation (Co-IP). After immunoprecipitation with anti-GFP agarose beads, precipitated proteins were probed with anti-Flag (upper panel) and anti-GFP (lower panel) antibody, separately. Images are representative of two independent experiments. **c** LCI assay showing that the association between TaPP2C-7A and TaPYL-1D was significantly repressed by TN1. Quantification of the relative luminescence intensities were calculated, the values in combination 3 were defined as "1". Data are means ± SEM ($n = 13$ biologically independent samples). Different letters indicate significant differences between groups, as determined by one way ANOVA with Duncan's multiple range tests ($p < 0.01$). **d** RT-qPCR assays indicating the relative expression levels of *TN1* and *tn1* in the infiltrated *N. benthamiana* leaf areas shown in (c). The data were normalized to *NbACT1*. Data are means ± SD of three biological replicates, and *p* values are indicated by two-tailed unpaired *t*-test. **e** A proposed working model for TN1 in the regulation of wheat tiller number. Source data are provided as a Source Data file.

and *Aegilops tauschii*[56]. The Arabidopsis *nced3-2* mutant displays enhanced bud outgrowth[20,21,57], and ABA-deficient mutants in rice also have more tillers[19]. In this study, we demonstrated that exogenous ABA treatment efficiently repressed tiller bud outgrowth in wheat, causing a low-tillering phenotype similar to that of the *tn1* mutant (Supplementary Fig. 17a). Consistently, the expression of *TaNCED3* genes and ABA levels increased in the *tn1* mutant (Fig. 5d, e); overexpression of *TaNCED3* could reduce the tiller numbers of these transgenic wheat lines (Supplementary Fig. 16). Moreover, we found that TN1 might repress ABA signaling by inhibiting the physical association between TaPYL and TaPP2C (Fig. 6c). Based on these findings, we propose a working model on TN1 in regulating wheat tillering (Fig. 6e): TN1 promotes wheat tillering, at least partially, through two layers of molecular mechanisms via repressing ABA biosynthesis and inhibiting ABA signaling through preventing the binding of TaPYL to TaPP2C.

In addition, TB1/BRC1 proteins play important roles in regulating tillering/branching in plants[18,32–34]. However, the expression pattern of *TaTB1* was not affected in the *tn1* mutant (Supplementary Fig. 13a, b), suggesting that TN1 regulates wheat tillering independently of the TB1 pathway.

## Methods

### Plant materials and growth conditions
The low-tillering *tn1* mutant was obtained from an EMS-mutagenized population of the wheat variety YZ4110[58]. An F₂ population consisting of 431 individuals derived from the cross between YZ4110 and the *tn1*

mutant was used for genetic analysis. The $F_2$ segregating population of 11,155 plants used in map-based cloning was generated by crossing *tn1* and JM20, a Chinese elite cultivar with strong tillering ability and high yield. The phenotype of $F_2$ individuals was further validated in the $F_3$ generation, and each line contained at least 10 individuals. All $F_2$ and $F_3$ individuals were hand-sown in 2-m rows with 30-cm row spacing, leaving ~14 cm between each plant, at the experimental fields of the Institute of Crop Sciences, Chinese Academy of Agricultural Sciences (CAAS).

The low-tillering line Fielder$^{tn1}$ and the free-tillering line Fielder$^{TN1}$ were identified from a BC$_4$F$_3$ population derived from the cross between *tn1* and Fielder. The $T_0$ generation transgenic plants were grown in pots (20 cm × 20 cm) in a growth chamber with 16 h light/8 h dark (25 °C/18 °C, 65% relative humidity), and the $T_1$ generation transgenic plants were grown in the experimental field of the Institute of Crop Sciences, CAAS, Beijing.

## Phenotypic and genetic analysis
The tiller numbers of $F_1$, $F_2$, and $F_3$ generations from different combinations were measured before harvesting. The frequency distributions of tiller number were determined, and the $\chi^2$ test was performed with Statistical Analysis Software (SAS, release 8, SAS Institute Inc.). Plant height, spike length, spikelet number, and grain number per spike were evaluated from at least 30 plants. The statistical analyses were performed with SPSS v20 (IBM); all the boxplots and bar charts were plotted by GraphPad Prism 8 (GraphPad Software).

## Bulked segregant analysis and wheat 660 K SNP array analysis
Thirty low-tillering and thirty high-tillering individuals with extreme phenotypes from the JM20 × *tn1* $F_{2:3}$ populations were prepared to construct wild-type pools and mutant-type pools. Genomic DNA was extracted by means of the CTAB method[59] and combined into one bulk sample in an equal ratio. The DNA quantity and quality were measured by a NanoDrop 2000 spectrophotometer (Thermo Fisher Scientific, USA) and gel electrophoresis.

The bulked samples were genotyped with the Axiom® Wheat 660 K SNP array using the Affymetrix GeneTitan® system based on the Axiom® 2.0 Assay Manual workflow protocol. Axiom™ Analysis Suit 3.0 software was used to perform SNP calling; polymorphic SNP markers were included in further analysis, and monomorphic and poor-quality SNP markers were excluded. The sequences of all probes were used to query IWGSC RefSeq v0.4 (https://urgi.versailles.inra.fr/blast/) by BLAST to obtain flanking sequences for marker design.

## KASP markers–based genotyping
Based on the flanking sequences of the probes from the Wheat 660 K SNP array and the genomic information of IWGSC RefSeq v0.4, the polymorphic SNPs between the wild-type pool and the mutant-type pool were selected and converted to KASP markers. The forward primers, carrying the standard FAM compatible tail (5′-GAAGGTGAC-CAAGTTCATGCT-3′) or HEX tail (5′-GAAGGTCGGAGTCAACGGATT-3′), contained the polymorphism between the wild-type and mutant-type pools at the 3′ end. The common reverse primers were carefully selected to enhance their specificity across the A, B, and D genomes. The sequences of KASP markers are listed in Supplementary Data 1.

The KASP assays were performed in 384-well plates according to protocol of LGC Genomics. Each reaction mixture consisted of a final volume of 5 μL containing 2.44 μL template DNA (40–80 ng), 2.5 μL V4.0 2× Master mix (LGC, Biosearch Technologies), 0.06 μL primer mix (12 μM each allele-specific primer and 30 μM common primer). The PCR reaction was performed on a QuantStudio™ 7 Flex instrument (Life Technologies Corporation, USA) under the following cycling conditions: denaturation at 95 °C for 15 min; 10 cycles of 95 °C for 20 s and touchdown starting at 62 °C for 1 min (decreased by 0.6 °C per cycle); and 30–42 cycles of amplification (95 °C for 20 s; 57 °C for

1 min). Data analysis was performed automatically by QuantStudio™ Real-Time PCR Software v1.3 (Applied Biosystems, USA).

## Histological analysis and in situ hybridization
Shoot bases were fixed in formalin–acetic acid–alcohol (50% ethanol, 5% glacial acetic acid, 3.7% formaldehyde, 35% DEPC-H$_2$O, v/v/v/v) at 4 °C overnight, dehydrated through a series of steps, and then embedded in paraffin (Paraplast Plus®, Sigma-Aldrich). A microtome (Leica RM2145) was used to slice the tissues into 8-μm sections, which were affixed to microscope slides. The slides were stained with 0.1% (w/v) toluidine blue O (Sigma, Germany) for 5–10 min and then observed and photographed through the ZEISS SteREO Discovery.V20 system (Germany).

The region from nucleotides 286 to 726 of the coding sequence of *TN1* gene was amplified by gene-specific primers with 5′-flanked T7 RNA polymerase promoter (5′-TAATACGACTCACTATAGGGTCTGCAG AGCAGCCGCAGGACT-3′ and 5′-TAATACGACTCACTATAGGGCAGCA GCTTCACCATGCGGCCG-3′), and then the sense and antisense RNA probes were in vitro transcribed by T7 RNA polymerase. Digoxigenin-labeled RNA probes were synthesized using a DIG northern Starter Kit (Roche, 11175025910) according to the manufacturer's instructions. RNA in situ hybridization with the antisense probe was performed on longitudinal sections of shoot bases. After blotting with Anti-digoxigenin AP-conjugate (Roche, 11093274910) and incubation with the NBT solution (Roche, 11383213001), the slides were observed and photographed through the ZEISS SteREO Discovery.V20 system (Germany).

## RNA extraction and quantitative RT-PCR analysis
Total RNA was extracted from various tissues using TRIzol™ Reagent (Invitrogen). First-strand cDNA synthesis was carried out using 5× All-In-One RT MasterMix (Applied Biological Materials Inc.) according to the manufacturer's instructions. The qPCR assays were performed on a Roche LightCycler® 96 thermal cycler instrument (Roche Applied Science) with *TaGAPDH* as the internal control. A SYBR® Premix Ex Taq Kit (TaKaRa, Japan) was used in a total volume of 15 μL with the following amplification program: 95 °C for 2 min, 40 cycles of 95 °C for 5 s, and 60 °C for 35 s. Each assay was performed in three biological replicates, and the quantitative variation was evaluated by the $2^{-\Delta\Delta CT}$ method[60]. The primers used in qPCR are listed in Supplementary Data 8.

## Vector construction and wheat transformation
To generate the vector for the functional complementation test, a 7357-bp genomic DNA fragment of *TN1* (including the 3251-bp putative promoter region upstream of the start codon, a 1840-bp gene body region, and a 2266-bp terminator region downstream of the stop codon) was amplified from YZ4110 genomic DNA and inserted into the binary vector pCAMBIA3301 to construct *TN1pro:TN1*. The resulting vector was introduced into Agrobacterium strain EHA105 and transformed into Fielder$^{tn1}$.

To generate the vector for gene editing via CRISPR/Cas9, two sgRNAs were designed in the 5′ UTR adjacent to the start codon and the first exon of *TN1* using the E-CRISP website[61]. The target sites (target1: 5′-AAGCTAGATAGGAACAGGA-3′; target2: 5′-CGTCGCGCCAC AAGGGTAC-3′) were incorporated into forward and reverse primers, respectively, and the PCR fragment amplified from the intermediate vector pCBC-MT1T2 was inserted into the binary vector pBUE413 using the 'Golden Gate' method[62,63]. The plasmid was transformed into the free-tillering wheat cultivar Fielder. The gene-editing vector with the target (5′-AGCCGTGGCCCAAGGTGTC-3′) for *TaNCED3* genes was constructed with the method described above and transformed into Fielder$^{tn1}$. For *TaNCED3-5D* (*TraesCS5D02G038800*) overexpression, the coding sequence was amplified and cloned into the binary vector pCAMBIA3301 driven by the maize *Ubiquitin* promoter, and was transformed into Fielder.

The tiller numbers of $T_1$ generation transgenic plants were measured for statistical analysis at the grain-filling stage. All transgenic wheat plants were obtained by a proprietary method for Agrobacterium-mediated transformation developed by the Plant Innovation Center, Japan Tobacco Inc., Iwata, Japan[64]. The transgenic plants were identified by PCR amplification and sequencing; all primers used for the construction preparation and the transgenic plant identification are listed in Supplementary Data 8.

## Promoter activity test and subcellular localization assay

The plasmids used in the promoter activity test and subcellular localization assay were constructed in pAN580-GFP, which is a pUC-based expression vector that includes the cauliflower mosaic virus *35S* (*CaMV35S*) promoter, the *GFP* reporter gene, and the *Nopaline synthase* (*NOS*) terminator. To generate the *TN1pro:GFP* construct, the 3251-bp putative promoter of the *TN1* gene was amplified and subcloned into the pAN580-GFP vector to replace the *CaMV35S* promoter. To create the *35Spro:TN1-GFP* construct, the full-length *TN1* coding sequence was amplified and inserted into the pAN580-GFP vector. The primers for these constructs are listed in Supplementary Data 8. To indicate the plasma membrane and endoplasmic reticulum, the *35Spro:PIP2-mCherry* construct was co-expressed during the subcellular localization experiment of TN1. These plasmids were transfected into wheat leaf mesophyll protoplasts by PEG-mediated methods[65]. After 16 h of incubation in the dark at 25 °C, the GFP and mCherry fluorescence was detected at excitation wavelengths of 488 and 543 nm, respectively, with a confocal laser scanning microscope (Carl Zeiss, LSM880).

## Phylogenetic analysis

The polypeptide sequence of the third ANK domain (33 amino acids) in TN1 was used as query to perform BLASTP searches against NCBI nonredundant protein databases with default parameters, retrieving 84 sequences from 22 species. The sequence with the best hit in each species was selected for the following analysis (Supplementary Data 3). Considering that only two sequences were from dicots among the 22 sequences, another BLASTP search was conducted using EnsemblPlants databases (http://plants.ensembl.org/index.html). Two other sequences (Cla97C09G163880 and Csa_3G457650) were obtained from the dicot plants *Citrullus lanatus* and *Camelina sativa*. Thus, 24 sequences (including TN1) were used for phylogenetic analysis (Supplementary Fig. 11a). Multiple alignments of the deduced amino acid sequences were performed using ClustalW with default parameters. The phylogenetic tree was constructed by MEGA 7.0 using the neighbor-joining method with 1000 bootstrap replicates. The sequence logo was obtained by Weblogo software[66].

## Transcriptome analysis

Shoot bases with nonelongated buds and the 2–5-cm tiller buds as shown in Fig. 5a, b were collected from 15–20 individuals. Total RNA was extracted as described above. RNA quantity and integrity were assessed with a NanoDrop 2000 (Thermo Scientific, Waltham, MA) and a Bioanalyzer 2100 (Agilent Technologies, Santa Clara, CA). The barcoded cDNA libraries were constructed using a TruSeq™ RNA Library Prep Kit v2 (Illumina, San Diego, CA) following the manufacturer's instructions. Then, the libraries were amplified through PCR, quantified by Qubit 2.0 (Thermo Scientific, Waltham, MA), and sequenced on an Illumina HiSeq 4000 platform; ~17 Gb raw data were obtained for each library. Adaptors and low-quality reads were trimmed by Fastp (v0.12.4) with default parameter[67]. Kallisto was used to calculate gene read counts and transcripts per million (TPM) base on the reference genome IWGSC RefSeq v1.0 and gene annotation v1.1[27,68]. Differential expression analysis was performed with edgeR[69] and genes with $\log_2$(fold-change) $\geq 1$ and *p* value < 0.05 were considered as DEGs. GO enrichment analysis was conducted using agriGO[70].

## Measurement of endogenous ABA and IAA levels

Approximately 100 mg (fresh weight) shoot bases and tiller buds of YZ4110 and the *tn1* mutant were homogenized in liquid nitrogen, weighed, and extracted for 24 h with methanol containing 2.5 ng $^2H_6$-ABA and $^2H_2$-IAA as internal standards. Endogenous ABA and IAA were then purified and measured[71,72]. Liquid chromatography (LC) with tandem mass spectrometry analysis was performed on a UPLC system (Waters) coupled to the 6500 Qtrap system (AB SCIEX). LC separation used a BEH C18 column (1.7 μm, 100 × 2.1 mm; Waters) with mobile phase A, 0.05% (v/v) acetic acid in water, and B, 0.05% (v/v) acetic acid in acetonitrile. The gradient was set with initial 20% B and increased to 70% B within 6 min. ABA and IAA were detected in multiple reaction monitoring mode with transition, and the transitions were as follows: ABA 263.0 > 153.1, $^2H_6$-ABA 269.2 > 159.2, IAA 174.0 > 130.0 and $^2H_2$-IAA 176.0 > 132.0. Three biological replicates were analyzed for each treatment.

## Chemical treatments of wheat seedlings

Stock solutions of 10 mM ABA (Sigma-Aldrich, USA) and 25 mM sodium tungstate (Sangon Biotech, Shanghai) were prepared in ethanol and water, respectively, and diluted with nutrient solution to the final concentrations. A 0.1% (v/v) ethanol solution or water was used as control treatments. The ABA and sodium tungstate treatments were conducted in a growth chamber with 16 h light/8 h dark (25 °C/18 °C, 65% relative humidity).

For the ABA treatment, seeds of YZ4110 and the *tn1* mutant were planted in 8 × 8 × 9.5-cm pots filled with vermiculite. Final ABA concentrations of 0, 5, 10, and 20 μM were added into the pots at the 7 d after germination, and the corresponding treatment was repeated in 3-d intervals. After three rounds of treatment, the seedlings were grown for another week.

For the sodium tungstate treatments, the seedlings were grown hydroponically in 11.5 × 7.5 × 8-cm boxes filled with nutrient solution for 7 d after germination. Final concentrations of 0, 5, 10, and 20 μM sodium tungstate were used for the treatment. After 1 h of treatment, the seedlings were transferred into fresh nutrient solution without sodium tungstate. The treatments were performed in 3-d intervals, after three rounds of treatment, and the seedlings were grown for another week.

## Dual-luciferase reporter assays

The ~2.7-kb DNA fragments of the promoter region of *TaNCED3-5B/5D* were amplified from YZ4110 and cloned into pGreenII 0800 vector to generate reporter constructs. The effector constructs *35Spro:TabZIP-5A/5B/5D* and the negative control *35Spro:YFP* were co-transfected with the reporter constructs in appropriate pairs. Different constructs combinations were transformed into *N. benthamiana* leaf mesophyll protoplasts by the PEG-mediated method[65]. After 18-h incubation in the dark at 25 °C, the activities of firefly luciferase (LUC) and *Renilla* luciferase (REN) were separately determined using a Dual-Luciferase Reporter Assay System (Promega, E1910, USA). The ratios of LUC to REN were calculated as the relative activity of the tested promoters. Primers are listed in Supplementary Data 8.

## LCI assay

In the LCI assay, the full-length *TN1* and *TaPP2C-7A* were cloned into the vector pCambia-cLUC, respectively, and *TaPYL-1D* was cloned into vector pCambia-nLUC. Then these constructs as well as the empty vectors (EV) were transformed into Agrobacterium strain GV3101 for further infiltration in *N. benthamiana* leaves[73]. Luciferase activities in LCI assay were measured after 36–48 h later using CCD-imaging apparatus (Berthold, LB985). Each assay was repeated at least three times. Primers are listed in Supplementary Data 8.

## Co-IP assay

The ankyrin repeat domain of TN1 (TN1$^{ANK}$, residues 1–422) was used for Co-IP assay. Constructs of target genes fused with different markers (TN1$^{ANK}$-GFP, TaPYL-1D-3×Flag) were verified by DNA sequencing and transformed into Agrobacterium strain GV3101, different combinations were prepared for further infiltration in *N. benthamiana* leaves. The Co-IP assays were carried out as published protocol[44]. In brief, total proteins were extracted and incubated with 30 μL GFP-Trap® magnetic agarose beads (Chromotek, gtma-20) following the manufacturers procedure. The TN1$^{ANK}$-GFP and GFP proteins were detected with anti-GFP (Roche, 11814460001, 1:2000 dilution), and TaPYL-1D-3×Flag were detected with anti-Flag (Sigma-Aldrich, A9044, 1:5000 dilution). The Goat anti-Rabit IgG-HRP (Abmart, M21002, 1:8000 dilution) was used to recognize the primary antibody. Primers were listed in Supplementary Data 8.

## Reporting summary

Further information on research design is available in the Nature Portfolio Reporting Summary linked to this article.

## Data availability

All data supporting the findings of this work are available within the paper and its Supplementary Information files. The datasets and plant materials generated and analyzed during the current study are available from the corresponding author upon request. The RNA-seq data have been uploaded to NCBI under BioProject ID PRJNA786783. Source data are provided with this paper.

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

## Acknowledgements

The authors would like to thank Prof. Qixin Sun and Zhongfu Ni of China Agricultural University for kindly providing the CRISPR/Cas9 vector system; Xueyong Zhang and Chenyang Hao of the Institute of Crop Sciences, CAAS, for checking the variation of *TN1* in the 145 wheat cultivars. We are grateful to Yongqiang Gu from USDA-ARS, Western Regional Research Center, Cheng Zou from Cornell University, Haiyang Wang from the South China Agricultural University, and Zefu Lu from the Institute of Crop Sciences, CAAS for their valuable suggestions in this study. We are also grateful to Lingli Zheng, Yuhong Liu, Lei Pan, Wuman Xu and Hao Cheng for their excellent technical support. We also thank Zhongxu Chen and Hao Liu from Chengdu Tcuni Inc. for their valuable

and efficient work on the RNA-seq experiment. This work was supported by the National Natural Science Foundation of China Grant (91935304 and 31991213) to X.K., the Central Public-interest Scientific Institution Basal Research Fund (S2022ZD02) to L.Z., Talent Program and Agricultural Science and the Technology Innovation Program of CAAS to X.K. and X.L.

## Author contributions

X.L., X.K., J.S., and J.J. designed the experiments and directed the research. C.D., L.Z., Q.Z., D.L., Z.X., G.C., Y.C., L.W., Z.L., G.L., X.Z., and C.X. performed the experiments. Y.Y. analyzed the RNA-seq data. C.L. and J.C. quantified the ABA contents. G.Z. gave the direct instructions in the utilization of the Chinese Spring genomic sequence. C.D., X.K., J.S., and L.Z. wrote the manuscript. All authors read and approved the paper.

## Competing interests

The authors declare no competing interests.
