## [Peer Review File · Nature Communications]

Tiller Number1 encodes an ankyrin repeat protein that controls tillering in bread wheatReviewers' Comments:

Reviewer #1:

Remarks to the Author:

The manuscript by Dong and colleagues entitled "Tiller Number 1 encodes an ankyrin repeat protein that controls tillering in bread wheat" reported cloning and functional study of Tiller Number 1 (TN1) in bread wheat. As tillering is one of the most important agricultural traits determining grain yield, it is of significant importance to clone key genes regulating tiller number and to elucidate molecular mechanisms underlying tillering regulation. The manuscript is well-written, and the findings are novel to the field of tillering regulation in wheat. But part of the conclusions is overstated and needs more evidence.

The following are points need to be further addressed:

1. The RNA-seq analysis and ABA content in tiller buds of WT and tn1 are key data of this study. As shown in lines 239-241 and 462-465, RNA sequencing analysis were performed using tiller bases of YZ4110 and tn1 at the third-leaf stage. As shown in Figure 1f, tillers of YZ4110 and tn1 showed dramatic difference in developmental stages. Because ABA biosynthesis and responsive genes are mainly expressed in dormant buds other than in elongated buds. The observation that several ABA biosynthesis genes and ABA-responsive genes were significantly up-regulated in the tn1 mutant was probably due to the obvious difference in growth stage. The authors should re-analyze these data and avoid possible misleading.
2. The authors measured endogenous levels of ABA and IAA using tiller buds of YZ4110 and tn1. Because ABA level in unelongated buds is higher than that in elongated buds, and the bud elongation is dramatically suppressed in tn1. The authors are suggested to clearly show tiller bud morphologies of YZ4110 and tn1 and to measure ABA levels using tiller buds of YZ4110 and tn1 at similar developmental stage.
3. Data in this study indicated that the reduced tiller number of tn1 is closely associated with increased ABA accumulation in the tiller buds, but cannot demonstrated that "TN1 promotes tiller bud outgrowth by repressing the expression of ABA biosynthesis genes and ABA accumulation in the tiller buds" as mentioned in lines 70-72. The authors are required to provide genetic evidence to support the importance of ABA accumulation in TN1-regulated tiller bud elongation.

Reviewer #2:

Remarks to the Author:

TN1 is shown to control an ankyrin repeat protein that controls tillering in bread wheat by mapping, complementation analysis and CRISPR. It is suggested tn1 may suppress tiller growth via regulation of ABA levels in buds. TN was shown to encode a membrane located protein and is expressed in shoot apical meristems and buds.

Questions/comments to address:

Tillering is an important determinant of yield in wheat. How does the tn1 mutant differ from mutants corresponding for tillering and yield QTLs in wheat? Or how do we know if this new line may be specific enough (non-pleiotropic enough) to enhance yield. If this is not known, this aspect should be at least deemphasised.

Sequencing in mutant and wild type buds showed 3848 differentially expressed genes. Given this large number it is important to understand the significance of the ABA related genes in particular. Has ABA been shown in bread wheat to inhibit tillering? Does ABA treatment to WT buds (axillary and apical – where the gene is expressed) phenocopy the mutant? Will an ABA synthesis or signalling inhibitor

revert the mutant phenotype back to wild type? Might the changes ABA levels be considerably downstream of other more direct changes pertinent to TN1?

As tiller bases (1cm in size) were used for gene expression and not specifically, axillary buds, it may be difficult to identify changes in BRC1 expression. Were buds used specifically for TB1 gene expression?

Reviewer #3:

Remarks to the Author:

This manuscript describes mutant generation and characterization by gene cloning and expression and phenotypic analysis for the wheat TN1 gene. The genetics and gene cloning are well executed and thorough and clearly demonstrate identification of the TN1 gene by complementation and CRISPR/Cas9 approaches. Gene cloning is becoming more straightforward in wheat with current technologies but still requires extensive effort, as showcased in this manuscript. The authors further show the TN1 gene product is an ankyrin-repeat protein that is membrane-localized. TN1 gene expression is enriched in vegetative and inflorescence shoot branches, consistent with a direct role in controlling bud outgrowth. The data for in situ RNA localization to tiller buds are acceptable given the difficulty of these experiments. Transcriptomics comparing wt and tn1 mutant reveal complex gene expression changes, consistent with the phenotype differences. The finding of ABA-biosynthetic gene expression changes is perhaps predictable from recent work in Arabidopsis and rice, but important to ascertain independently in wheat and a key finding of this work. Notably, ABA accumulation is increased in mutant tiller buds, consistent with NCED3 gene expression changes. Overall, except for the phylogenetic analyses the work is high quality and the conclusions are well justified and supported by the data. The work incrementally advances our understanding of the tillering process in plants by extending to wheat the evidence for a role of ABA in regulating tillering and plant architecture. These results will be of general interest to a broad range of plant biologists and to those interested in the plant biotechnology.

Major points:

- Functional evidence of a molecular mechanism that more directly or indirectly links the TN1 gene product to NCED3 expression levels would greatly strengthen the manuscript. For example, among the transcription factors showing expression changes, might any directly regulate the NCED3 genes?
- The relatively high expression levels of TN1 in young spike begs the question of whether or not tn1 mutants have an inflorescence branching defect.
- The manuscript requires editing throughout for correct English.

Minor points:

- ABA feeding experiments would strengthen the manuscript, to determine if addition of ABA inhibits tiller outgrowth in wt wheat.

line 134 "We detected no differences in other genes between YZ4110 and the tn1 mutant" Do they mean for genes in the interval; if so for how many? Please clarify, the reader should not have to figure out how many genes were examined by looking the the primers used for PCR.

lines 140-142 Usage of the word "similar" here is potentially misleading. To support the gene cloning the genotypes must match exactly, not just be similar. Please clarify.

line 150 The language here is confusing and misleading, to incorrectly indicate that transformation was used to create the Fielder-tn1 line. The Methods section is much clearer.

line 221 The phylogenetic analysis is weak. The phylogeny (Supp Fig 10a) does not provide clear support the authors have identified orthologs or paralogs of TN1 for the comparison. Even within the grasses most subclades are species-specific or only include genes from wheat and barley. Given the brevity of the text and no explanation in the Methods for these analyses, it is not clear of the authors have selected genes closely related to TN1 (as desired) or simply, for example, some ankyrin-repeat genes or other, more distant homologs. While the motivation is justified to examine amino acid conservation at the sites mutated in tn1, the data are not well explained or convincing.

Point-by-point response to Reviewers:

Reviewer #1 (Remarks to the Author):

The manuscript by Dong and colleagues entitled "Tiller Number 1 encodes an ankyrin repeat protein that controls tillering in bread wheat" reported cloning and functional study of Tiller Number 1 (TN1) in bread wheat. As tillering is one of the most important agricultural traits determining grain yield, it is of significant importance to clone key genes regulating tiller number and to elucidate molecular mechanisms underlying tillering regulation. The manuscript is well-written, and the findings are novel to the field of tillering regulation in wheat. But part of the conclusions is overstated and needs more evidence.

The following are points need to be further addressed:

1. The RNA-seq analysis and ABA content in tiller buds of WT and *tn1* are key data of this study. As shown in lines 239-241 and 462-465, RNA sequencing analysis were performed using tiller bases of YZ4110 and *tn1* at the third-leaf stage. As shown in Figure 1f, tillers of YZ4110 and *tn1* showed dramatic difference in developmental stages. Because ABA biosynthesis and responsive genes are mainly expressed in dormant buds other than in elongated buds. The observation that several ABA biosynthesis genes and ABA-responsive genes were significantly up-regulated in the *tn1* mutant was probably due to the obvious difference in growth stage. The authors should re-analyze these data and avoid possible misleading.

Response:

Thank you very much for your suggestions. In the revised manuscript, we re-performed and re-analyzed the RNA-seq data using the shoot bases with unelongated tiller buds and the 2–5 cm tiller buds (Fig. 5a, b), respectively. The results showed that the key ABA biosynthetic genes (*TaNCED3-5B/5D*) and ABA signaling-related genes (*TaPYL*, *TaPP2C* and *TaSnRK2*) were significantly up-regulated in both the shoot base and the elongated tiller bud, respectively, in the *tn1* mutant (Fig. 5c, d, Supplementary Fig. 13).

2. The authors measured endogenous levels of ABA and IAA using tiller buds of YZ4110 and *tn1*. Because ABA level in unelongated buds is higher than that in elongated buds, and the bud elongation is dramatically suppressed in *tn1*. The authors are suggested to clearly show tiller bud morphologies of YZ4110 and *tn1* and to measure ABA levels using tiller buds of YZ4110 and *tn1* at similar developmental stage.

Response:

In the revised version, the ABA/IAA measurements were re-performed using the unelongated tiller buds and the elongated tiller buds, respectively. The morphologies of the shoot base and the tiller bud used for ABA/IAA measurements and gene expression analyses were shown in Fig. 5a–b. The results showed that the ABA levels were increased in both the unelongated tiller buds and the elongated tiller buds in the *tn1* mutant compared with YZ4110 (Fig. 5e). However, the IAA levels were not altered in both the unelongated tiller buds and the elongated tiller buds in the *tn1* mutant compared with YZ4110 (Fig. 5f).

3. Data in this study indicated that the reduced tiller number of *tn1* is closely associated with

increased ABA accumulation in the tiller buds, but cannot demonstrated that “TN1 promotes tiller bud outgrowth by repressing the expression of ABA biosynthesis genes and ABA accumulation in the tiller buds” as mentioned in lines 70-72. The authors are required to provide genetic evidence to support the importance of ABA accumulation in TN1-regulated tiller bud elongation.

Response:

To answer the reviewer’s concerns, we performed the exogenous ABA feeding experiments and demonstrated that exogenous ABA treatment could efficiently repress the tiller bud outgrowth in YZ4110, causing a phenotype similar to the low-tillering phenotype of *tn1* mutant (Supplementary Fig. 15a). On the other hand, the sodium tungstate (Na_2WO_4 , an ABA synthesis inhibitor) treatment could partially rescue the low-tillering phenotype of *tn1* mutant (Supplementary Fig. 15b).

These results suggest that the low-tillering phenotype of *tn1* mutant is likely caused by the increased ABA levels in the tiller bud. We now changed the sentence “TN1 promotes tiller bud outgrowth by repressing the expression of ABA biosynthesis genes and ABA accumulation in the tiller buds” to “We demonstrated that the inhibition of tiller bud outgrowth in the *tn1* mutant may be caused by the enhanced ABA accumulation in the tiller bud” in Lines 70–72.

Reviewer #2 (Remarks to the Author):

TN1 is shown to control an ankyrin repeat protein that controls tillering in bread wheat by mapping, complementation analysis and CRISPR. It is suggested *tn1* may suppress tiller growth via regulation of ABA levels in buds. TN was shown to encode a membrane located protein and is expressed in shoot apical meristems and buds.

Questions/comments to address:

1. Tillering is an important determinant of yield in wheat. How does the *tn1* mutant differ from mutants corresponding for tillering and yield QTLs in wheat? Or how do we know if this new line may be specific enough (non-pleiotropic enough) to enhance yield. If this is not known, this aspect should be at least deemphasised.

Response:

Currently, we still do not know how to manipulate the *TN1* gene in increasing wheat yield. Therefore, we deemphasized the aspect on wheat yield in the revised manuscript.

2. Sequencing in mutant and wild type buds showed 3848 differentially expressed genes. Given this large number it is important to understand the significance of the ABA related genes in particular. Has ABA been shown in bread wheat to inhibit tillering? Does ABA treatment to WT buds (axillary and apical – where the gene is expressed) phenocopy the mutant? Will an ABA synthesis or signalling inhibitor revert the mutant phenotype back to wild type? Might the changes ABA levels be considerably downstream of other more direct changes pertinent to TN1?

Response:

We appreciated the reviewer’s suggestion. Previous studies have shown that exogenous ABA could repress the tiller bud development in both rice and wheat (Liu et al., 2020; Yu et al., 2021).

In the revised manuscript, we performed the exogenous ABA feeding experiments and demonstrated that exogenous ABA treatment could efficiently repress the tiller bud outgrowth in YZ4110, causing

a phenotype similar to the low-tillering phenotype of *tn1* mutant (Supplementary Fig. 15a). Treatment with the ABA synthesis inhibitor (sodium tungstate) partially rescued the low-tillering phenotype of *tn1* mutant (Supplementary Fig. 15b). These results suggest that the low-tillering phenotype of *tn1* mutant is likely caused by the increased ABA levels in the tiller bud.

Moreover, we found that the expression levels of TabZIP-5A/5B/5D transcription factors, the homologs of maize *ZmbZIP4* that could directly activate the expression of *ZmNCED3* (Ma et al., 2018), were increased in the *tn1* mutant (Supplementary Fig. 14a). We also demonstrated that the TabZIP-5A/5B/5D transcription factors could activate the expression of *TaNCED3* genes in the transient transactivation system (Supplementary Fig. 14b–d).

Reference

- Liu X, et al. ζ -Carotene isomerase suppresses tillering in rice through the coordinated biosynthesis of strigolactone and abscisic acid. *Mol. Plant* **13**, 1784-1801 (2020).
- Yu H, et al. Regulation of 2,4-D isooctyl ester on *Triticum aestivum* and *Aegilops tauschii* tillering and endogenous phytohormonal responses. *Front. Plant Sci.* **12**, 642701 (2021).
- Ma H, et al. *ZmbZIP4* contributes to stress resistance in maize by regulating ABA synthesis and root development. *Plant Physiol.* **178**, 753-770 (2018).

3. As tiller bases (1cm in size) were used for gene expression and not specifically, axillary buds, it may be difficult to identify changes in BRC1 expression. Were buds used specifically for TB1 gene expression?

Response:

Thank you for pointing out this issue. In the revised manuscript, we used the shoot base and the tiller bud (Fig. 5a, b) for RNA-seq and qRT-PCR analyses. The results showed that the expression levels of *TaTB1-4A/4B/4D* genes did not show obvious differences between YZ4110 and the *tn1* mutant (Supplementary Fig. 12a, b).

Reviewer #3 (Remarks to the Author):

This manuscript describes mutant generation and characterization by gene cloning and expression and phenotypic analysis for the wheat TN1 gene. The genetics and gene cloning are well executed and thorough and clearly demonstrate identification of the TN1 gene by complementation and CRISPR/Cas9 approaches. Gene cloning is becoming more straightforward in wheat with current technologies but still requires extensive effort, as showcased in this manuscript. The authors further show the TN1 gene product is an ankyrin-repeat protein that is membrane-localized. TN1 gene expression is enriched in vegetative and inflorescence shoot branches, consistent with a direct role in controlling bud outgrowth. The data for in situ RNA localization to tiller buds are acceptable given the difficulty of these experiments. Transcriptomics comparing wt and *tn1* mutant reveal complex gene expression changes, consistent with the phenotype differences. The finding of ABA-biosynthetic gene expression changes is perhaps predictable from recent work in Arabidopsis and rice, but important to ascertain independently in wheat and a key finding of this work. Notably, ABA accumulation is increased in mutant tiller buds, consistent with *NCED3* gene expression changes. Overall, except for the phylogenetic analyses the work is high quality and the conclusions are well justified and supported by the data. The work incrementally advances our understanding of the tillering process in plants by extending to wheat the evidence for a role of ABA in regulating

tillering and plant architecture. These results will be of general interest to a broad range of plant biologists and to those interested in the plant biotechnology.

Major points:

1. Functional evidence of a molecular mechanism that more directly or indirectly links the TN1 gene product to NCED3 expression levels would greatly strengthen the manuscript. For example, among the transcription factors showing expression changes, might any directly regulate the NCED3 genes?

Response:

Thanks for your suggestions. We noticed that the expression levels of TabZIP-5A/5B/5D transcription factors, the homologs of maize ZmbZIP4 that could directly activate the expression of *ZmNCED3* (Ma et al., 2018), were increased in the *tn1* mutant (Supplementary Fig. 14a). Moreover, the TabZIP-5A/5B/5D transcription factors could activate the expression of *TaNCED3* genes in the transient transactivation system (Supplementary Fig. 14b–d).

Reference

Ma H, et al. ZmbZIP4 contributes to stress resistance in maize by regulating ABA synthesis and root development. *Plant Physiol.* **178**, 753-770 (2018).

2. The relatively high expression levels of TN1 in young spike begs the question of whether or not *tn1* mutants have an inflorescence branching defect.

Response:

Indeed, we observed that the inflorescence development was affected in the *tn1* mutant as shown in the figure below (quantification of development according to Waddington et al., 1983). The spike length and spikelet number were reduced in the *tn1* mutant compared with YZ4110 (Supplementary Fig. 1b, c).

Inflorescence morphologies of YZ4110 and the *tn1* mutant

W3, Glume primordium present; **W3.5**, Floret primordium present; **W4**, Stamen primordium present; **W5.5**, Stylar canal closing; ovarian cavity enclosed on all sides but still open above; **W10**, Styles curved outwards and stigmatic branches spread wide; pollen grains on well-developed stigmatic hairs.

Reference

Waddington SR, Cartwright PM, Wall PC. A quantitative scale of spike initial and pistil development in barley and wheat. *Ann. Bot.* **51**, 119-130 (1983).

3. The manuscript requires editing throughout for correct English.

Response:

We have carefully edited the manuscript again.

Minor points:

1. ABA feeding experiments would strengthen the manuscript, to determine if addition of ABA inhibits tiller outgrowth in wt wheat.

Response:

Thanks very much for your valuable suggestions. In the revised manuscript, we demonstrated that exogenous ABA treatment could efficiently repress the tiller bud outgrowth in YZ4110, causing a phenotype similar to the low-tillering phenotype of *tn1* mutant (Supplementary Fig. 15a). On the other hand, the ABA synthesis inhibitor (sodium tungstate) treatment could partially rescue the low-tillering phenotype of *tn1* mutant (Supplementary Fig. 15b). These results suggest that the low-tillering phenotype of *tn1* mutant is likely caused by the increased ABA levels in the tiller bud.

2. line 134 "We detected no differences in other genes between YZ4110 and the *tn1* mutant" Do they mean for genes in the interval; if so for how many? Please clarify, the reader should not have to figure out how many genes were examined by looking the the primers used for PCR.

Response:

Thanks for your suggestions. We modified this part in Lines 130–135 in the revised version.

3. lines 140-142 Usage of the word "similar" here is potentially misleading. To support the gene cloning the genotypes must match exactly, not just be similar. Please clarify.

Response:

Thanks for your suggestions. We modified this part in Lines 140–143 in the revised version.

4. line 150 The language here is confusing and misleading, to incorrectly indicate that transformation was used to create the Fielder-*tn1* line. The Methods section is much clearer.

Response:

Thanks for your suggestions. We modified this part Lines 148–151 in the revised version.

5. line 221 The phylogenetic analysis is weak. The phylogeny (Supp Fig 10a) does not provide clear support the authors have identified orthologs or paralogs of TN1 for the comparison. Even within the grasses most subclades are species-specific or only include genes from wheat and barley. Given the brevity of the text and no explanation in the Methods for these analyses, it is not clear of the authors have selected genes closely related to TN1 (as desired) or simply, for example, some ankyrin-repeat genes or other, more distant homologs. While the motivation is justified to examine amino acid conservation at the sites mutated in *tn1*, the data are not well explained or convincing.

Response:

Thanks very much for your suggestions. In the revised version, we focused on examining the amino acid conservation at the mutation sites of the third ANK domain (33 amino acids) in the TN1 protein.

We performed BLASTP to search against NCBI non-redundant protein databases using the third ANK domain as bait, 84 sequences from 22 species were obtained and the sequence with the best

hit in each species were selected for the following analysis (Supplementary Data 1). Considering that only two sequences were from dicots among the 22 sequences, we conducted another BLASTP search using EnsemblPlants databases. Two other sequences (Cla97C09G163880, Csa_3G457650) were obtained from dicot plants *Citrullus lanatus* and *Camelina sativa*, sharing the 50.0% and 48.3% sequence identities with the third ANK domain of TN1, respectively. Thus, totally 24 sequences (including TN1) were used for phylogenetic analysis (Supplementary Data 1, Supplementary Fig. 10a). We found the amino acid Ala125 is highly conserved in those plants (Supplementary Fig. 10b). For detail information, please see the revised manuscript.

Reviewers' Comments:

Reviewer #1:

Remarks to the Author:

In this revised manuscript, the authors have performed useful additional experiments and addressed many of the specific experimental criticisms raised by myself and other reviewers. However, the genetic analysis and novelty of this research need improve.

The following points need to be further addressed:

1. The authors have generated transgenic lines overexpressing TaNCED3-5D, and mentioned that transgenic lines showed reduced tiller number (Supplementary Fig. 15a, b). They also generated knockout lines of various TaNCED3 genes in the Fieldertn1 background via CRISPR/Cas9 and observed their tiller number. The reviewer notes that statistical analyses of tiller number have not been provided, leading to the doubt that the tiller number of each line was based on phenotypes of the T0 generation or the T1/T2 generation. It is inaccurate to compare the tiller number of T0 transgenic line that was differentiated from transgenic calli with the tiller number of WT (FielderTN1) and mutant (Fieldertn1) grown from seeds. It is also difficult to guarantee whether a specific independent transgenic line was differentiated from an independent transgenic callus. Based on this general role of genetic analysis, the authors are strongly suggested to make statistical analysis of tiller number using stable T1 or T2 transgenic plants.

2. Another important point is about the novelty of this research. I agree with other reviewers that the roles of ABA in suppressing bud elongation have been shown in Arabidopsis, rice and wheat, and NCEDs have been identified as the key enzyme regulated by upstream transcription factors (OsHOX12, ZmNCED3) in the signaling pathway. The authors showed several expected results and identified a similar mechanism in wheat without providing enough novel discovery.

3. TN1 has been proven to be colocalized with PIP2-mCherry in the plasma membrane (PM) and endoplasmic reticulum (ER). It is interesting to investigate how a PM-ER-localized TN1 regulates the expression of nuclear-encoded genes, such as TaZIP-5, TaPYL and TaSnRK2. This will strengthen the novelty of this research.

Reviewer #2:

Remarks to the Author:

Outstanding issues remaining include:

The study still has not addressed whether the changes in expression of ABA genes or ABA level are causal or consequential to the phenotype.

Buds were already grown out in WT compared to mutants at time of hormone and gene expression measurement – so it is not possible to distinguish cause from effect.

Supp 15; NCED-OE lines need to provide a mean and standard error as well as information on the generation, T1, T2 etc? Expression levels in what tissue? Sodium tungstate is not an ABA specific inhibitor.

Reviewer #3:

Remarks to the Author:

The manuscript from Dong et al is a resubmission and includes additional data. Overall the manuscript

is strengthened, especially the expression analysis and the more careful selection of appropriate wt and mutant tissues, and direct evidence of ABA effects on tillering to accompany the mutant analysis. Regarding my specific comments and the authors' responses:

Point 1: Additional data regarding potential direct regulators of the NCED3 genes are provided. Further clarification of the mechanism of NCED3 expression changes appears outside the scope of the current study.

Point 2: queried whether or not tn1 mutants show altered branching in the inflorescence shoot, as they do in the vegetative shoot, given that TN1 is expressed highly in the inflorescence. Data relevant to this point are provided in the response to reviewers though those data are not included in the manuscript. These images indicate slower overall spike growth as the developmental basis for mature morphology differences: generally smaller inflorescences in tn1 mutants. Contrary to the text of the response, however, there is apparently no branching defect in tn1. The images of spike development are not included in the revised ms, nor is the interpretation I stated above, so while this response clarifies the point to me, the reviewer, the information is not in the paper or transmitted to the reader.

Point 3: The English is better in the revision.

The minor points raised in my review were all addressed adequately.

Point-by-point response to Reviewers:

Reviewer #1 (Remarks to the Author):

In this revised manuscript, the authors have performed useful additional experiments and addressed many of the specific experimental criticisms raised by myself and other reviewers. However, the genetic analysis and novelty of this research need improve.

The following points need to be further addressed:

1. The authors have generated transgenic lines overexpressing *TaNCED3-5D*, and mentioned that transgenic lines showed reduced tiller number (Supplementary Fig. 15a, b). They also generated knockout lines of various *TaNCED3* genes in the *Fielder^{ml}* background via CRISPR/Cas9 and observed their tiller number. The reviewer notes that statistical analyses of tiller number have not been provided, leading to the doubt that the tiller number of each line was based on phenotypes of the T0 generation or the T1/T2 generation. It is inaccurate to compare the tiller number of T0 transgenic line that was differentiated from transgenic calli with the tiller number of WT (*Fielder^{TNI}*) and mutant (*Fielder^{ml}*) grown from seeds. It is also difficult to guarantee whether a specific independent transgenic line was differentiated from an independent transgenic callus. Based on this general role of genetic analysis, the authors are strongly suggested to make statistical analysis of tiller number using stable T₁ or T₂ transgenic plants.

Response: Thank you very much for pointing this out. In the revised manuscript, we performed statistical analyses for the tiller numbers of the stable transgenic wheat plants in the T₁ generation. We also showed that the T₁ generation *TaNCED3-5D* overexpressing lines displayed significantly reduced tiller numbers compared with *Fielder* plants (Supplementary Fig. 16a–c). As for the T₁ generation knockout lines of *TaNCED3* genes in the *Fielder^{ml}* background, the sextuple mutants of *TaNCED3* genes displayed partially rescued tiller numbers compared with *Fielder^{ml}* (*Tanced3-c 2#–5#*, Supplementary Fig. 18a–c), while the triple mutant showed no obvious difference compared with *Fielder^{ml}* (*Tanced3-c 1#*, Supplementary Fig. 18a–c).

2. Another important point is about the novelty of this research. I agree with other reviewers that the roles of ABA in suppressing bud elongation have been shown in *Arabidopsis*, rice and wheat, and *NCEDs* have been identified as the key enzyme regulated by upstream transcription factors (OsHOX12, ZmNCED3) in the signaling pathway. The authors showed several expected results and identified a similar mechanism in wheat without providing enough novel discovery.

Response: Thank you for your comments and concerns. In our view, the main novelty of this study is the cloning of a novel wheat tillering regulatory gene *TNI*, which encodes a

transmembrane ankyrin repeat (ANK-TM) protein. Another novelty of this study is the underlying mechanism of TN1 in promoting wheat tiller bud outgrowth through two layers of repression on ABA pathway: inhibits ABA biosynthesis via repressing the transcription of *TaNCED3s* genes and inhibiting ABA signaling through preventing the binding of ABA receptor TaPYL to TaPP2C (Fig. 6).

3. TN1 has been proven to be colocalized with PIP2-mCherry in the plasma membrane (PM) and endoplasmic reticulum (ER). It is interesting to investigate how a PM-ER-localized TN1 regulates the expression of nuclear-encoded genes, such as TaZIP-5, TaPYL and TaSnRK2. This will strengthen the novelty of this research.

Response: Thank you very much for your valuable suggestions. In the revised manuscript, we demonstrated that TN1 could physically interact with TaPYL-1D, the plasma membrane receptor of ABA (Fig.6a, b). We found that TN1 but not tn1 could inhibit the physical association between TaPYL and TaPP2C. Therefore, we propose that TN1 might participate in the ABA signaling pathway, at least partly, through inhibiting the physical association between TaPYL and TaPP2C (Fig. 6e).

Reviewer #2 (Remarks to the Author):

Outstanding issues remaining include:

1. The study still has not addressed whether the changes in expression of ABA genes or ABA level are causal or consequential to the phenotype.

Response: Thank you for pointing this out. Based on the experimental results available, we think that the changes in expression of ABA genes and ABA level may be the causal to the phenotype. Firstly, exogenous ABA treatment efficiently represses tiller bud outgrowth in YZ4110, causing a phenotype similar to the low-tillering phenotype of the *tn1* mutant (Supplementary Fig. 17a), and the treatment with the ABA biosynthesis inhibitor sodium tungstate partially rescued the low-tillering phenotype of the *tn1* mutant (Supplementary Fig. 17b). Secondly, we knocked out the *TaNCED3* homologous genes in the Fielder^{*tn1*} background; the sextuple mutants of *TaNCED3* genes from in the T₁ generation displayed partially rescued tiller numbers compared with Fielder^{*tn1*} (*Tanced3-c 2#~5#*, Supplementary Fig. 18a–c).

Taken together, these evidences support our view that the changes in expression of ABA genes and ABA level, at least partially, may be the cause of the phenotype.

2. Buds were already grown out in WT compared to mutants at time of hormone and gene expression measurement – so it is not possible to distinguish cause from effect.

Response: In the revised version, we measured the ABA levels again using the shoot base and elongated tiller buds, respectively. The ABA levels were obviously increased in both the shoot base and tiller buds of *tn1* mutants compared with the WT (Fig. 5a, b).

3. Supp 15; NCED-OE lines need to provide a mean and standard error as well as information on the generation, T1, T2 etc? Expression levels in what tissue? Sodium tungstate is not an ABA specific inhibitor.

Response: Thank you very much for pointing out these issues. In the revised manuscript, we developed the T₁ generation NCED-OE lines and made statistical analysis of tiller numbers (Supplementary Fig. 16a, c). In addition, the expression levels were measured using the flag leaf tissue at heading stage. Indeed, sodium tungstate is not an ABA specific inhibitor. It is a potent inhibitor of molybdo-enzymes in plants such as ABA aldehyde oxidase (Martin-Rodriguez JA et al., 2011). Therefore, sodium tungstate could inhibit the biosynthesis of ABA, and treatment with sodium tungstate partially rescued the low-tillering phenotype of the *tn1* mutant (Supplementary Fig. 17b). We also used another ABA synthesis inhibitor, fluridone (Sigma-Aldrich, USA), to do the chemical treatment assays. Fluridone is a kind of herbicide that blocks phytoene desaturase in the carotenoid synthetic pathway (Bartels and Watson, 1978). Because the carotenoids are precursors of ABA, Fluridone is also an inhibitor of ABA biosynthesis, but the negative effect of fluridone is the extensive damage to the plastid. In our experiment, whether continuous treatment or short-term treatment (1 h treatment, only once), the leaf color of plant seedlings turned into pale (as shown in the figure below). As a result, sodium tungstate was selected to conduct further experiments.

Morphologies of YZ4110 and *tn1* mutant under fluridone treatment

The wheat seedlings at first leaf stage were hydroponically cultured in nutrient solution with fluridone for 1 h, and then grown at normal conditions.

References

Martin-Rodriguez JA, et al. Ethylene-dependent/ethylene-independent ABA regulation of tomato plants colonized by arbuscular mycorrhiza fungi. *New Phytol* **190**, 193-205 (2011).

Bartels PG, Watson CW. Inhibition of carotenoid synthesis by fluridone and norflurazon. *Weed Science* **26**, 198-203 (1978).

Reviewer #3 (Remarks to the Author):

The manuscript from Dong et al is a resubmission and includes additional data. Overall the manuscript is strengthened, especially the expression analysis and the more careful selection of appropriate wt and mutant tissues, and direct evidence of ABA effects on tillering to accompany the mutant analysis. Regarding my specific comments and the authors' responses:

1. Additional data regarding potential direct regulators of the NCED3 genes are provided. Further clarification of the mechanism of NCED3 expression changes appears outside the scope of the current study.

Response: Thank you very much for your comments.

2. Queried whether or not *tn1* mutants show altered branching in the inflorescence shoot, as they do in the vegetative shoot, given that TN1 is expressed highly in the inflorescence. Data relevant to this point are provided in the response to reviewers though those data are not included in the manuscript. These images indicate slower overall spike growth as the developmental basis for mature morphology differences: generally smaller inflorescences in *tn1* mutants. Contrary to the text of the response, however, there is apparently no branching defect in *tn1*. The images of spike development are not included in the revised ms, nor is the interpretation I stated above, so while this response clarifies the point to me, the reviewer, the information is not in the paper or transmitted to the reader.

Response: Thank you very much for pointing this out. In the revised version, we added the inflorescence morphologies of YZ4110 and the *tn1* mutant, as shown in the Supplementary Fig. 8.

3. The English is better in the revision.

The minor points raised in my review were all addressed adequately.

Response: Thank you again for your comments.

Reviewers' Comments:

Reviewer #1:

Remarks to the Author:

The revised manuscript by Dong and colleagues is much improved. The authors have performed useful experiments and added reasonable explanation. I am largely quite satisfied with the authors' response. I feel that this study is interesting and a nice contribution to the study of hormone biosynthesis and signaling in crops. I'd like to support its publication.

Reviewer #2:

Remarks to the Author:

no further comment

REVIEWERS' COMMENTS

Point-by-point response to Reviewers:

=====

Reviewer #1 (Remarks to the Author):

The revised manuscript by Dong and colleagues is much improved. The authors have performed useful experiments and added reasonable explanation. I am largely quite satisfied with the authors' response. I feel that this study is interesting and a nice contribution to the study of hormone biosynthesis and signaling in crops. I'd like to support its publication.

Response: Thank you again for your comments.

Reviewer #2 (Remarks to the Author):

no further comment

Response: Thank you very much.